 **eLIFE**

# Elba, a novel developmentally regulated chromatin boundary factor is a hetero-tripartite DNA binding complex

**Tsutomu Aoki[1], Ali Sarkeshik[2], John Yates[2], Paul Schedl[1]\***

[1]Department of Molecular Biology, Princeton University, Princeton, United States;
[2]Department of Chemical Physiology, Scripps Research Institute, La Jolla, United States

**Abstract** Chromatin boundaries subdivide eukaryotic chromosomes into functionally autonomous domains of genetic activity. This subdivision insulates genes and/or regulatory elements within a domain from promiscuous interactions with nearby domains. While it was previously assumed that the chromosomal domain landscape is fixed, there is now growing evidence that the landscape may be subject to tissue and stage specific regulation. Here we report the isolation and characterization of a novel developmentally restricted boundary factor, Elba. We show that Elba is an unusual hetero-tripartite protein complex that requires all three proteins for DNA binding and insulator activity.

**\*For correspondence:**
pschedl@princeton.edu

**Competing interests:** The authors have declared that no competing interests exist

**Reviewing editor**: Jim Kadonaga, University of California-San Diego, United States

## Introduction

Eukaryotic chromosomes are subdivided into functionally autonomous domains by special elements called chromatin boundaries (*Gaszner and Felsenfeld, 2006*). Boundaries define units of independent genetic activity by shielding genes and/or regulatory elements within a domain from adventitious interactions with regulatory elements/genes in nearby domains. The specific functions that can be ascribed to most boundary elements include an enhancer-blocking or insulator activity (*Holdridge and Dorsett, 1991*; *Kellum and Schedl, 1991*; *Chung et al., 1993*), a silencer-blocking or barrier activity (*Kellum and Schedl, 1991*; *Chung et al., 1993*; *Burgess–Beusse et al., 2002*) and, when combined in specific pairwise combinations an ability to bring distant chromosomal DNA segments in close proximity (*Cai and Shen, 2001*; *Muravyova et al., 2001*). Boundary elements have a diverse array of functions. In the yeast *S. pombe*, boundaries help restrict and maintain the heterochromatin state of the silenced mating type locus (*Noma et al., 2001*). In *Drosophila*, boundary elements play a critical role in the Bithorax complex (BX-C), helping to ensure that the three BX-C homeotic genes properly specify segmental identity (*Maeda and Karch, 2006*). In vertebrates, boundaries have been implicated in controlling the expression of mRNAs encoding different neuronal Protocadherin-alpha isoforms and in regulating recombination and expression of immunoglobulin light and heavy chain genes (*Guo et al., 2011*; *Ribeiro de Almeida et al., 2011*; *Monahan et al., 2012*).

The first boundaries identified were able to block adventitious regulatory interactions irrespective of cell type or developmental stage (*Gyurkovics et al., 1990*; *Holdridge and Dorsett, 1991*; *Kellum and Schedl, 1991*; *Chung et al., 1993*). Moreover, the proteins conferring insulating activity in flies and vertebrates like Su(Hw), BEAF, Zw5, GAGA factor and CTCF are ubiquitously expressed and seemingly functional in all cell types. This led to the idea that boundaries are static structures and consequently that the regulatory domain landscape of eukaryotic chromosomes is largely invariant from one cell to the next. However recent studies argue that the domain landscape is more dynamic than previously imagined and that genes can be differentially regulated during normal development and differentiation and in the progression of disease states like cancer by redefining their domain

**eLife digest** If all of the DNA in a human cell was stretched out, it would be about 2 m long. The nucleus of a human cell, on the other hand, has a diameter of just 6 μm, so the DNA molecules that carry all the genetic information in the cell need to be carefully folded to fit inside the nucleus. Cells meet this challenge by combining their DNA molecules with proteins to form a compact and highly organized structure called chromatin. Packaging DNA into chromatin also reduces damage to it.

But what happens when the cell needs to express the genes carried by the DNA as proteins or other gene products? The answer is that the compact structure of chromatin relaxes and opens up, which allows the DNA to be transcribed into messenger RNA. Indeed, packing DNA into chromatin makes this process more reliable, thus ensuring that the cell only produces proteins and other gene products when it needs them. However, because cross-talk between neighboring genes could potentially disrupt or change gene expression patterns, cells evolved special elements called boundaries or insulators to stop this from happening. These elements subdivide eukaryotic chromosomes into functionally autonomous chromatin domains.

Since the protein factors implicated in boundary function seemed to be active in all tissues and cell types, it was assumed for many years that these boundaries and the resulting chromatin domains were fixed. However, a number of recent studies have shown that boundary activity can be subject to regulation, and thus chromatin domains are dynamic structures that can be defined and redefined during development to alter patterns of gene expression.

Aoki et al. report the isolation and characterization of a new fruit fly boundary factor that, unlike previously characterized factors, is active only during a specific stage of development. The Elba factor is also unusual in that it is made of three different proteins, known as Elba1, Elba2, and Elba3, and all three must be present for it to bind to DNA. While Elba2 is present during most stages of development, the other two Elba proteins are only present during early embryonic development, so the boundary factor is only active in early embryos. In addition to revealing a new mechanism for controlling boundary activity as an organism develops, the studies of Aoki et al. provide further evidence that chromatin domains can be dynamic.

organization (**Bell and Felsenfeld, 2000**; **Witcher and Emerson, 2009**; **Gomes and Espinosa, 2010**). For example, regulation of the imprinted *Igf2* and *H19* genes pivots on parent of origin differences in the DNA methylation pattern and consequently the activity of a CTCF-dependent boundary element upstream of the *H19* gene (**Bell and Felsenfeld, 2000**; **Hark et al., 2000**). In this and also other examples, regulatory domains are defined or redefined locally by modulating how a ubiquitous insulator protein functions at a specific boundary element. However, one mechanism for defining domains that has yet to be described are developmental stage or cell type specific changes in the available repertoire of boundary proteins.

In previous studies we discovered that the insulating activity of one of the *Drosophila* Bithorax (BX-C) complex boundaries, *Fab-7*, depends upon distinct, stage specific boundary factors (**Schweinsberg and Schedl, 2004**). Like other boundaries in BX-C, *Fab-7* is required to ensure the functional autonomy of its flanking parasegment specific *cis*-regulatory domains, *iab-6* (parasegment 11) and *iab-7* (parasegment 12) (**Gyurkovics et al., 1990**; **Figure 1**). When *Fab-7* is deleted the *iab-6* and *iab-7* domains no longer function independently and instead fuse into a single domain that incorrectly specifies parasegment 11. While the primary function of *Fab-7* in BX-C is to prevent inappropriate crosstalk between regulatory elements in *iab-6* and *iab-7*, it can also block interactions between enhancers/silencers and promoters both in the context of the endogenous BX-C (**Mihaly et al., 1997**) and in reporter gene assays (**Hagstrom et al., 1996**; see **Figure 1—figure supplement 1**).

Transgene assays and deletion mutations within BX-C have shown that the *Fab-7* boundary spans a DNA segment of 1.2 kb that contains two major nuclease hypersensitive sites, HS1 and HS2, and a minor hypersensitive site '*' (see **Figure 1B**). Like other well-characterized *Drosophila* boundaries, it has insulating activity throughout development apparently irrespective of tissue or cell type both in the context of BX-C and in transgene assays. However, *Fab-7* appears unusual in that its constitutive boundary function is generated by sub-elements whose activity is developmentally restricted. The first

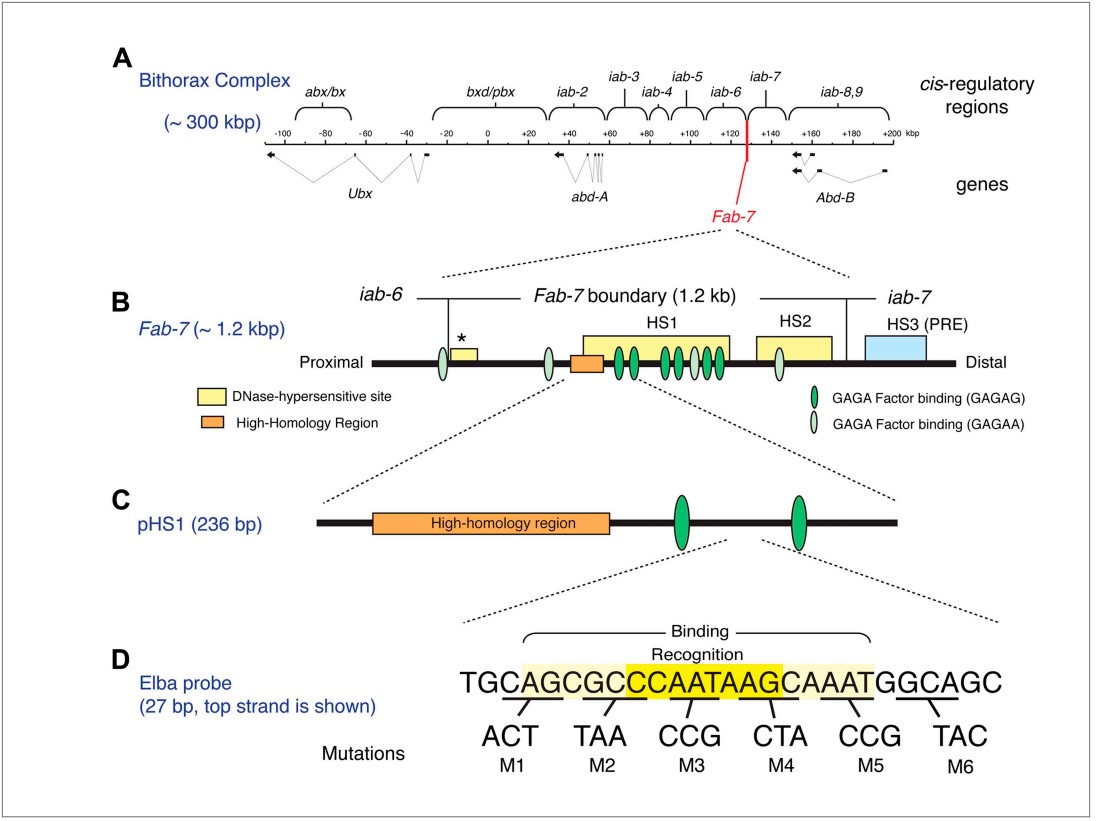

**Figure 1**. The Bithorax Complex, the *Fab-7* boundary and the Elba recognition element. (**A**) *Drosophila* Bithorax complex (BX-C). BX-C spans ~300 kb and includes three Hox-family genes *Ultrabithorax*, *abdominal-A* and *Abdominal-B*. Parasegment specific expression of these three homeotic genes is generated by a series of functionally autonomous *cis*-regulatory domains: *abx/bx*, *bxd/pbx* and *iab-2-iab8, 9*. Functionally autonomy depends upon boundary elements that lie between each *cis*-regulatory domain (***Maeda and Karch, 2010***). One of these boundary elements is *Fab-7*, which is located in between the *iab-6* and *iab-7 cis*-regulatory domains. Both in the context of BX-C and in transgene assays, the *Fab-7* boundary can block the action of enhancers/silencers at all stages of development, apparently irrespective of tissue or cell type (***Galloni et al., 1993***; ***Hagstrom et al., 1996***; ***Mihaly et al., 1997***; ***Schweinsberg et al., 2004***). (**B**) The *Fab-7* boundary spans a sequence of 1.2 kb and consists of two prominent and one minor (*) chromatin specific nuclease hypersensitive regions (shown as yellow boxes). There is a third prominent nuclease hypersensitive region (blue) just distal to the boundary, which corresponds to a Polycomb Response Element (PRE) for the *iab-7 cis*-regulatory domain (***Maeda and Karch, 2010***). The orange box is a ~100 base pair (bp) high-homology region which is conserved among *Drosophila* species (>90%) (***Aoki et al., 2008***). The ovals are binding sites for Trithorax-like (GAGA factor). (**C**) pHS1 is a 236-bp fragment from the proximal side of HS1 which has enhancer-blocking activity only in early embryos (***Schweinsberg and Schedl, 2004***). pHS1 includes the high-homology region and two GAGA-binding sites. These two GAGA sites are important for the early boundary activity of *Fab-7*, while GAGA sites elsewhere in *Fab-7* are needed later in development (***Schweinsberg et al., 2004***). In addition to the GAGA sites, the enhancer-blocking activity of pHS1 in early embryos also depends upon an 8-bp sequence, CCAATAAG, called Elba (Early boundary activity). Mutations in this sequence compromise the blocking activity of a 4×pHS1 multimer, while multimerization of a 27-bp oligo spanning the Elba sequence (8×Elba) [see (**D**)] is sufficient to confer early blocking activity. The Elba sequence is recognized by the stage-specific Elba DNA-binding factor. Elba factor binding is detected in 0–6 hr nuclear extracts, but it is absent in 6–12 hr (and 6–18 hr) nuclear extracts (***Aoki et al., 2008***). (**D**) Sequence of the 27-bp oligo used as the Elba probe in the EMSA experiments shown in ***Figures 3A, 4, 5B, and 6***. The Elba factor in 0–6 hr nuclear extracts recognizes the 8-bp Elba sequence (shaded by yellow) and requires an additional 5 bp both upstream and downstream for full binding activity (shaded by light yellow). The bases underlined were altered as indicated in the mutant oligos, M1–M6. These mutant oligos were used as cold competitors in ***Figures 4C*** and ***6C*** as indicated. For the DNA affinity beads, a 27-bp oligo containing the mutation M3 was used as the mutant Elba sequence.

The following figure supplements are available for figure 1.

**Figure supplement 1**. Enhancer blocking activity of Fab-7, pHS1×4 and the Elba×8 multimer.

evidence for this developmental restriction came from the effects of mutations in binding sites for the GAGA factor in enhancer blocking assays (*Schweinsberg et al., 2004*). As shown in *Figure 1B*, there are three pairs of GAGA sites in HS1. Mutations in the proximal pair weaken boundary activity in early embryos, but have little effect from mid-embryogenesis onwards. In contrast, mutations in the central pair weaken boundary activity during mid-embryogenesis and in adults, but not in early embryos. Further evidence for sub-elements with developmentally restricted boundary activity came from experiments in which small fragments from HS1 were multimerized. Multimers of small 2- to 400-bp fragments from the distal half of HS1 were found to block enhancer-promoter interactions from mid-embryogenesis onwards even more efficiently that the intact *Fab-7* boundary. However, these fragments have less blocking activity than *Fab-7* in early embryos. Conversely, a multimerized 236-bp fragment, pHS1, containing the proximal pair of GAGA sites has enhancer blocking activity during early embryogenesis, but not thereafter. This is illustrated in *Figure 1—figure supplement 1* which shows that pHS1×4 blocks the *fushi tarazu* (*ftz*) UPS stripe enhancer in early embryos, but doesn't block the *ftz* NE enhancer during mid-embryogenesis. Consistent with these transgene results, two partial *Fab-7* deletions in BX-C that remove the distal half of HS1 plus H2, but retain the proximal pHS1 sequence including pair of GAGA sites function as boundaries during early embryogenesis but not later in development (*Schweinsberg and Schedl, 2004*). In addition, the early boundary activity of these partial deletions depends upon the GAGA factor (*Schweinsberg et al., 2004*).

With the aim of identifying factors in addition to GAGA that confer the early boundary activity of pHS1, we used probes from pHS1 for EMSA (electrophoresis mobility shift assay) experiments with staged nuclear extracts. Only one DNA binding activity, Elba, had a stage-specificity consistent with the insulating activity of pHS1 (*Aoki et al., 2008*). It is present in extracts from early 0–6 hr embryos, a period when pHS1 boundary activity is high. In contrast, in extracts from older 6–12 hr embryos, where pHS1 boundary activity in vivo is largely absent, only little Elba is detected. Elba recognizes an asymmetric 8-bp sequence, CCAATAAG (see *Figure 1D*), which is conserved in the *Fab-7* elements of other *Drosophila* species including the distant *melanogaster* relative *D. virilis*. Moreover, two lines of evidence indicate that the Elba factor is important for boundary activity in early embryos. We found that mutations in the recognition sequence that disrupt Elba binding in nuclear extracts compromise pHS1 insulator activity in vivo. Conversely, multimerizing the Elba recognition sequence is sufficient to confer early insulating activity (*Aoki et al., 2008*; see *Figure 1—figure supplement 1*).

Understanding the role of Elba in the context of BX-C and more generally in the establishment of chromatin domains during early embryogenesis requires the identification and characterization of this novel boundary factor. We describe here a general cross-affinity purification strategy for identifying components of multi-protein DNA binding complexes by mass spectrometry. Using this purification strategy, we show that the Elba boundary factor is an unusual hetero-tripartite protein complex. Two of the proteins, Elba1 and Elba2, share a conserved C-terminal 'BEN domain' (*Abhiman et al., 2008*). The third protein, Elba3, has no obvious conserved protein domains, but is encoded by a gene that is closely linked to Elba1. All three Elba proteins are required to reconstitute DNA binding activity in vitro. DNA binding depends upon the BEN domains in Elba1 and Elba2, while Elba3 mediates the assembly of an active Elba complex by interacting with the N-terminal domains of Elba1 and Elba2. All three Elba proteins are present in the Elba complex detected in 0–6 hr nuclear extracts and all three are required for chromatin domain boundary function in vivo. Finally, because *elba1* and *elba3* are 'mid-blastula transition' genes, Elba only binds to its' target sequences in *Fab-7* and confers insulator activity during early embryogenesis.

## Results

### Isolation of the Elba factor

We tried to isolate Elba directly from nuclear extracts by DNA-affinity purification. However, excess non-specific DNA-binding activity in the extracts inhibited Elba binding to the affinity beads and necessitated the use of a pre-purification procedure (see *Figure 2*, 'Materials and methods'). Following S-Sepharose chromatography, the active fraction was split into two aliquots and fractionated on either wild type (WT) or mutant (M) DNA affinity beads. Since Elba was substantially enriched in the 1.0 M KCl fraction from WT beads (WE1), while there was little in the corresponding fraction from mutant beads (WE2) (not shown), we analyzed the protein composition by mass spectrometry. However, in two independent single DNA-affinity purifications it wasn't possible to identify the Elba factor as far

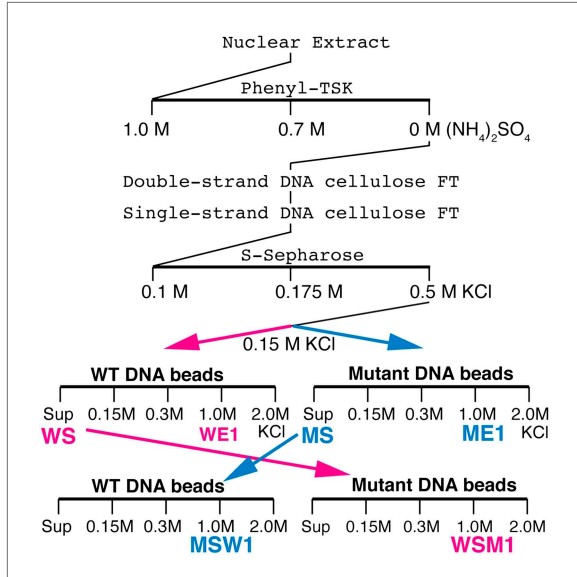

**Figure 2**. Elba purification scheme. Details provided in 'Materials and methods'.

too many proteins were detected in ME1 (and also ME1) and none stood out from the rest (see 'Materials and methods').

To reduce the number of remaining non-specific binding proteins and increase the differences in Elba yield between WT and M affinity beads, we devised the cross affinity purification scheme shown at the bottom of *Figure 2*. As before, the S-Sepharose faction was incubated with either WT or M affinity beads. However, instead of isolating Elba from the 1.0 M KCl eluates, we recovered it from the supernatant of WT (WS) and M (MS) affinity beads, respectively. As expected Elba was substantially depleted from WS, but enriched in MS (compare WS Input and MS Input in *Figure 3A*). We then fractionated WS on mutant affinity beads, and MS on WT affinity beads. *Figure 3A* shows that substantial amounts of Elba were recovered in the 1.0 M KCl fraction (MSW1) from WT beads, but not from mutant beads (WSM1).

The mass spectrometry dataset from the cross-affinity purification fractions was quite different from that of the conventional single-affinity purifications. First, the total number of proteins detected was greatly reduced, suggesting that many non-specific binding proteins were pre-cleared by the first step in the cross-affinity purification. Second, while there were still 176 unique proteins in the MSW1 fraction, three of these were substantially enriched compared to all others (*Table 1*). Each had 16 or more confirmed peptides with high spectral counts and their sequence coverage was over 25%. One protein, Elba1: CG12205, is encoded by a previously described mid-blastula transition gene, Bsg25A, of unknown function (*Singer and Lengyel, 1997*) while the other two, Elba2: CG9883 and Elba3: CG15634, are products of uncharacterized genes.

Although the three proteins have no previously known DNA binding domains, there are several intriguing connections. First, all three are ~40 kDa, which is close in size to the protein species in nuclear extracts that is UV cross-linked to probes containing the Elba sequence (*Aoki et al., 2008*). Second, the C-terminal ~130 amino acids of Elba1 and Elba2 show extensive sequence similarities (*Figure 3C*). Moreover, included within this region of similarity is a conserved ~90 amino acids BEN domain (*Figure 3B*) that has been implicated in protein:protein interactions and transcriptional regulation (*Cha et al., 1997*; *Mackler et al., 2000*; *Wang et al., 2006*; *Korutla et al., 2009*; *Duan et al., 2011*). Two other fly proteins have BEN domains. One is a predicted isoform of the Mod(mdg4)C boundary factor, while the other is Insensitive (Insv) (*Figure 3C*), which functions in neurogenesis and Notch signaling (*Duan et al., 2011*). Interestingly, the *elba2* and *Insv* transcription units are paired with each other (*Figure 7—figure supplement 1*). Mammals have a large family of BEND proteins. Of the mammalian proteins, Elba1 is most closely related to BEND7, while Elba2 is most closely related to BEND6 (*Figure 3D*). The BEN domain of Insv is most closely related to BEN9/hNAC2 and BEND8/hNAC1. Finally, though Elba3 differs from Elba1 and Elba2 in that it has no distinctive domains, its transcription unit is located next to *elba1* (*Figure 7—figure supplement 1*) just like the *elba2* and *insv* pair.

## Three different proteins are required to reconstitute Elba DNA binding activity in vitro

To determine if one of these highly enriched proteins corresponds to the Elba factor, each was synthesized by in vitro translation of the corresponding full length mRNAs and tested for DNA binding activity. *Figure 4A* shows that none shifted the Elba probe. We next translated all three mRNAs together. Strikingly, this combination generated a prominent shift that co-migrates with the Elba shift produced by 0–6 hr nuclear extracts (*Figure 4A*). To ascertain which proteins must be present to generate the Elba shift we in vitro translated each mRNA separately, and then mixed the translation

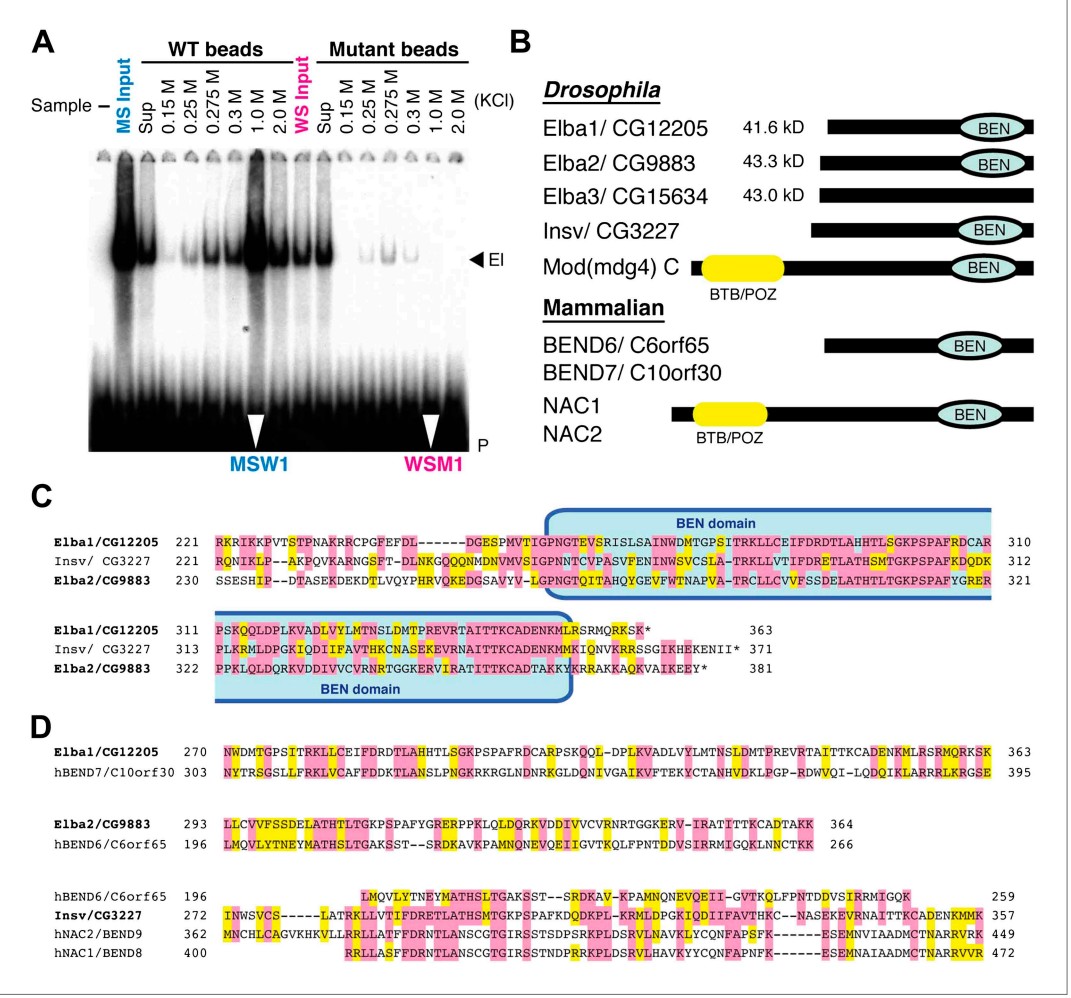

**Figure 3**. EMSA and Elba factor proteins. (**A**) EMSA of fractions from the cross-affinity purification. The $^{32}$P-labeled Elba probe was incubated with fractions as indicated and subjected to 4% acrylamide-gel electrophoresis. EI: Elba shift. P: probe. (**B**) Schematic of the Elba factors and BEN domain-containing (green) orthologs. BTB/POZ (yellow) domain is absent from Elba factors but is present in related proteins. (**C**) Sequence alignment of the C-terminal half of the *Drosophila* Elba1, Elba2 and a third *Drosophila* Ben protein Insv (Insensitive). The sequences of the three proteins were aligned according to the results of NCBI (National Center for Biotechnology Information) blast search (bl2seq). The amino acid residues conserved in more than two proteins are shaded with red. The residues that have similarities with each other are shaded with yellow. The predicted BEN domain region is boxed with pale blue. (**D**) Sequence alignment of *Drosophila* and mammalian (human) orthologs of BEN domain proteins. The C-terminal sequences of Elba1, Elba2 and Insv were subjected to blast search with human databases. Within the BEN domain sequences, the closest human ortholog of Elba1 is BEND7 (BEN domain-containing 7)/C10orf30 (29% identical, 46% positive), whereas for Elba2 the closest ortholog is BEND6/C6orf65 (29% identical, 53% positive). The BEN domain of Insv is most similar to three proteins: NAC2 (Nucleus accumbens-associated protein 2, NACC2)/BEND9, BEND6 and NAC1 (Nucleus accumbens-associated protein 1, NACC1)/BEND9. The amino acid residues conserved with each *Drosophila* protein are shaded with red. The residues that have similarities with each *Drosophila* protein are shaded with yellow.

reactions in all pairwise combinations. *Figure 4B* shows that none of the pairwise combinations had DNA binding activity; however, it was possible to reconstitute the Elba shift by combining all three translation products.

These findings demonstrate that all three Elba proteins are required to reconstitute an Elba-like DNA binding activity in vitro. To confirm that the properties of the reconstituted Elba factor match the endogenous factor in 0–6 hr nuclear extracts, we tested for sequence specificity. We previously identified the core recognition sequence by introducing a series of 3-bp point mutations into the Elba

**Table 1.** List of proteins unique to the 1.0 M KCl fraction from wild type DNA affinity beads (MSW1) in the third, cross-affinity, Elba purification

|  | Hit protein | Sequence count | Spectrum count | Sequence coverage | Mol. wt. |
|---|---|---|---|---|---|
| 1 | CG12205 (Bsg25A)-PA/gil7295685* | 23 | 372 | 56.5%/49%* | 41,583/47,368* |
| 2 | CG15634-PA | 16 | 150 | 32.20% | 43,008 |
| 3 | CG9883-PA | 17 | 63 | 26.80% | 43,310 |
| 4 | CG12052 (lola) 17 subtypes | 3 | 21 | 10.8–5.2% | 49,320–98,162 |
| 5 | CG12052 (lola)-PY | 3 | 21 | 8.70% | 62,746 |
| 6 | CG12052 (lola), unknown subtype | 3 | 21 | 5.10% | 107,303 |
| 7 | CG12052 (lola)-PD, -PE | 3 | 21 | 6.70% | 79,439 |
| 8 | CG14339-PA | 2 | 17 | 2.00% | 117,908 |
| 9 | CG2368(psq)-PD, -PE, -PF, PG, -PH/-PB, -PC | 6 | 15 | 12.7%/7.7% | 70,298/114,984 |
| 10 | CG1249 (snRNP2)-PA | 3 | 13 | 31.90% | 13,504 |
| 11 | CG6944 (Lam)-PA, -PB, -PC | 8 | 12 | 19.10% | 71,249 |
| 12 | CG16973 (msn)-PE | 4 | 11 | 4.50% | 115,386 |
| 13 | CG16973 (msn)-PC, -PD/-PB /-PA | 4 | 11 | 4.3%/3.9%/3.1% | 120,610/130,341/162,378 |
| 14 | CG11700 (CR11700)-PA | 2 | 11 | 10.60% | 34,335 |
| 15 | Reverse_CG31284 | 2 | 11 | 3.10% | 110,298 |
| 16 | CG3561 (KH1)-PA | 8 | 10 | 15.30% | 59,597 |
| 17 | CG10067 (Act57B)-PA/CG7478 (Act79B)-PA/CG18290 (Act87E)-PA, -PB | 5 | 10 | 14.40% | 41,835/41,787/41,802 |
| 18 | CG5178 (Act88F)-PA | 5 | 10 | 14.40% | 41,700 |
| 19 | CG1759 (cad)-PA, -PB | 4 | 9 | 4.40% | 51,306 |
| 20 | CG12154 (oc)-PA | 3 | 9 | 5.20% | 69,666 |
| 21 | CG6801 (l(3)j2D3)-PA | 8 | 8 | 19.20% | 44,830 |
| 22 | CG18124 (mTTF)-PA | 7 | 8 | 16.30% | 48,281 |
| 23 | CG7583 (CtBP)-PD, -PB, -PC, -PA | 6 | 8 | 20.50% | 42,252 |
| 24 | CG13634-PA | 2 | 8 | 4.30% | 52,594 |
| 25 | CG3143 (foxo)-PC, -PB | 5 | 7 | 9.60% | 67,413 |

*An alternative transcript of CG12205 (gi:7295685) that encoded a slightly larger protein was listed in the Genbank database at the time of this experiment. However, that sequence was subsequently removed from the CG12205 sequence list.

In the third, cross-affinity purification, there were 176 proteins in the MSW1 fraction (see *Figure 2*) which were not present in the WSM1 fraction. These 176 proteins were sorted according to the descending order of 'Spectrum Count' numbers. The 25 proteins with the highest 'Spectrum Count' are shown here. The top three proteins have a high spectrum count, and also have a high sequence count.

probe (*Aoki et al., 2008*; see *Figure 1D*). We used the same set of competitors to compare the in vitro reconstituted Elba factor with the endogenous factor in 0–6 hr nuclear extracts. *Figure 4C* shows that sequence specificity of reconstituted Elba is the same as nuclear extract Elba. Like the endogenous factor, the binding activity of reconstituted Elba is competed by excess amounts of wild type and M1, M5 and M6 DNA, whereas M2, M3 and M4 compete poorly or not at all.

## All three Elba proteins are components of the Elba factor in 0–6 hr nuclear extracts

The in vitro reconstitution experiments indicate that Elba is a complex composed of three distinct proteins, Elba1, Elba2 and Elba3. To determine if these three proteins are also components of the

endogenous nuclear factor, we generated two independent polyclonal antibodies against each Elba protein. We then tested the effects of the immune and corresponding pre-immune sera on the Elba shift in nuclear extracts. *Figure 4D* shows that all six immune sera give an Elba supershift, while the Elba shift is unchanged by the corresponding pre-immune sera. As an additional control for specificity we tested whether these sera would supershift probes from elsewhere in *Fab-7* that are recognized by factors which are present in 6–12 hr but not 0–6 hr nuclear extracts. We found that they did not. These findings indicate that the Elba factor in 0–6 hr nuclear extracts must also contain all three Elba proteins. While these results do not exclude the possibility that the Elba factor in nuclear extracts contains additional proteins besides Elba1–3, such proteins do not remain stably associated with Elba during its purification nor are they required for its DNA binding activity in vitro.

## Characterization of the Elba complex

As hetero-tripartite DNA binding factors are unusual and there are no previously known DNA binding domains in the Elba proteins, we sought to learn more about the organization and DNA binding activities of the Elba complex. In preliminary experiments, we found very weak, but detectable DNA binding activity when we mixed bacterially expressed N-terminal GST (Glutahione-S-transferase) fusions of the two BEN domain proteins, Elba1 and Elba2, in the absence of Elba3. Since GST is known to form stable dimers, we reasoned that the GST moieties might be substituting for Elba3 by promoting the formation of GST-Elba1:GST-Elba2 heterodimers (*Figure 6—figure supplement 2*). This finding together with the asymmetric Elba binding motif, CCAATAAG, suggested a model in which the BEN domains in Elba1 and Elba2 are responsible for DNA recognition, while the function of Elba3 is to bring Elba1 and Elba2 together so that they can form the DNA binding pocket (*Figure 5C*).

To test this model, we deleted the BEN domain from the Elba1 and Elba2 cDNAs (*Figure 5A*) and determined whether the resulting in vitro translated ΔBEN proteins can reconstitute DNA binding activity. *Figure 5B* shows that mixing the three full-length proteins (which have N-terminal FLAG tags as do all of the other proteins in *Figure 5A*) generates the Elba shift, while the shift is eliminated whenever a BEN domain deletion is used instead for reconstitution.

We next determined whether the combination of Elba1 and Elba2 is sufficient to reconstitute DNA binding if they are provided with heterologous GST dimerization domains. To maximize the formation of heterodimers, we co-translated mRNAs encoding the GST-Elba1F and GST-Elba2F fusions and assayed for DNA binding activity. As shown in lane 6 of *Figure 6A*, a prominent shift is generated when these two GST fusions are co-translated. In contrast, the individual GST fusions fail to shift the Elba probe on their own even though they are expected to form homodimers (*Figure 6A*, lanes 7 and 8). Consistent with the idea that only the heterodimer is able to bind to the Elba probe, we found that there is little DNA binding activity when the two GST fusions are translated independently and then mixed together just before assaying (not shown). In this case, DNA binding activity is likely limited by a slow rate of dissociation of the GST-ElbaF homodimers. The fact that a heterologous dimerization domain can support what appears to be near full DNA binding activity would be consistent with the idea that the role of Elba3 is to link Elba1 and Elba2 together (*Figure 6—figure supplement 1*). While GST can substitute for Elba3, the supershift generated by the addition of Elba3 (*Figure 6A*, lane 5) indicates that the GST-Elba1F:GST-Elba2F heterodimer is still able to form a trimeric complex with Elba3.

We entertained the possibility that the BEN deletions don't remove DNA binding domains, but instead eliminate sequences used by Elba3 for linking Elba1 and Elba2. In this case, the GST moiety should substitute for the BEN domain. However, this is not the case. When the BEN domain was deleted from either one or both of the GST-fused Elba1 and Elba2 proteins (ΔBEN), DNA-binding activity was completely abolished (*Figure 6A*, lanes 9–11).

To further characterize the functional domains in the three Elba proteins we subdivided them into N-terminal and C-terminal halves (*Figure 5A*). As expected from the experiments described above, the N-terminal halves of Elba1 and Elba2 had no DNA binding activity either with (*Figure 6B*) or without the GST moiety. For the C-terminal halves of Elba1 and Elba2 we first generated cDNAs that span the entire 130 amino acids homology region (including the 90 amino acids BEN domain) either with or without the N-terminal GST (*Figure 5A*). Like the full-length GST fusions, the GST-ElbaC homodimers have no detectable DNA binding activity on their own (*Figure 6A*, lanes 14 and 15), while a very prominent shift is generated when the RNAs encoding GST-Elba1C and GST-Elba2C are co-translated (lane 13). In addition to having roughly equivalent activity to the 'native' hetero-tripartite Elba complex

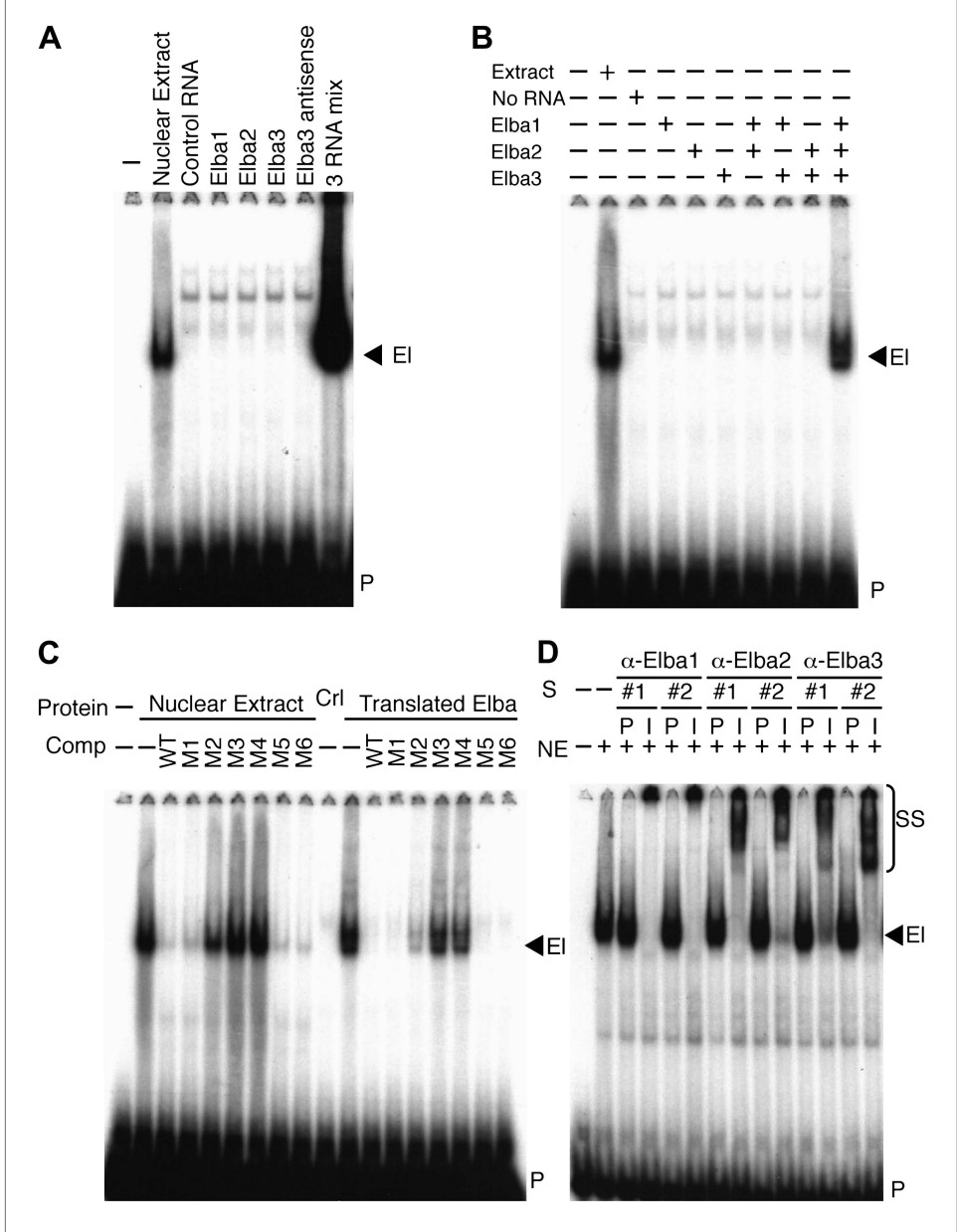

**Figure 4**. Elba is a hetero-tripartite complex. (**A**) and (**B**) All three Elba proteins are required to reconstitute DNA binding activity. In vitro translated proteins either singly or in combination as indicated were incubated with the Elba probe. El: Elba shift. P: probe. (**C**) Reconstituted Elba has the same sequence specificity as nuclear Elba. Reconstituted and nuclear extract shifts with or without (minus) a 100-fold excess of competitor (Comp) as indicated above each lane. WT: wild-type probe. M1–M6: mutant probes (see **Figure 1D**). Ctl: no-RNA control. (**D**) Nuclear Elba factor has all three Elba proteins. Nuclear extracts (NE) incubated with preimmune (P) or immune (I) polyclonal rabbit serum as indicated. #1, #2: serum from different rabbits. SS: antibody supershifts. El: Elba shift. P: free probe.

generated by the three full length Elba proteins, the GST-Elba1C:GST-Elba2C heterodimer exhibits a similar sequence specificity (**Figures 4C** and **6C**). The two ElbaC proteins also resemble the corresponding full-length Elba1 and Elba2 proteins in that full DNA binding activity requires the dimerization activity of the GST (compare **Figure 6A**, lanes 13 16). On the other hand, the two ElbaC proteins differ from the full length Elba1 and Elba2 proteins in two respects. Most importantly, the addition of Elba3 to the two ElbaC proteins (without the GST moiety) does not reconstitute full DNA binding activity (**Figure 6D**, lane 7). This finding suggests that Elba3 orchestrates the formation of the

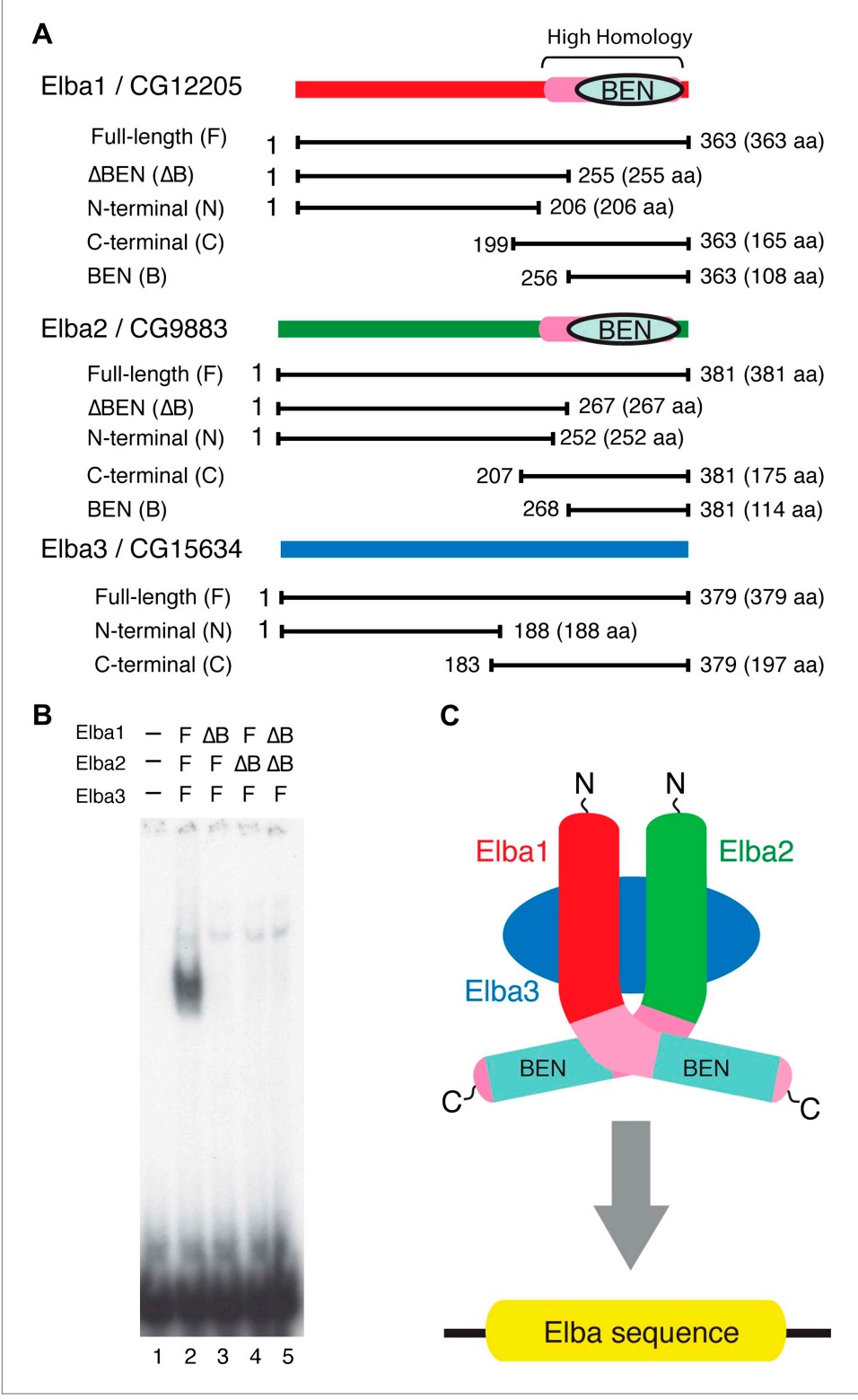

**Figure 5**. Elba protein deletions and functional organization of the Elba complex. (**A**) Elba proteins and deletion mutants. The bars under each diagram indicate the sequences retained in the mutant protein. The numbers correspond to the amino acid residues at the N and C terminal ends of the protein. The letter in parentheses is used to designate proteins added to the lanes in panel B and in ***Figure 6***. Each protein was expressed either with an N-terminal FLAG tag or with an N-terminal FLAG tag plus a GST tag. The FLAG tag was used to determine the relative amount of each protein so that the input of the translated proteins in the gel shift experiments could be
*Figure 5. Continued on next page*

*Figure 5. Continued*
adjusted. (**B**) BEN domain is required for the DNA-binding activity of Elba. EMSA experiments were performed as described in *Figure 4* by translating the full-length (F) or ΔBEN (ΔB) mutant Elba subunits in vitro and by mixing each as indicated above the lanes. (**C**) A schematic structure of proposed model for the Elba complex.

hetero-tripartite complex by interacting with sequences in the N-terminal domains of Elba1 and Elba2 (*Figure 5C*). Consistent with this idea, the shift generated by the active GST-Elba1C:GST-Elba2C heterodimer differs from the full-length heterodimer in that it is not supershifted by the addition of Elba3 (*Figure 6A*, lane 12). A second potentially interesting difference is that the mixture of Elba1C and Elba2C proteins (lacking GST) gives a faint shift that can be detected in long (*Figure 6D*, lane 8 filled arrowhead) but not short (*Figure 6A*, lane 16) exposures. Like the shift generated by the corresponding GST proteins, this shift is not enhanced or supershifted by the addition of Elba3 (lane 7) and it requires both proteins (*Figure 6D*, lanes 9 and 10). Interestingly, it can be stabilized as well as supershifted by the addition of FLAG antibodies (but not GST antibodies: lanes 11 and 12). Although further studies will be required, it would appear that there are weakly active hetero-dimerization motifs in the C-terminal halves of Elba1 and Elba2 that become more readily accessible when the N-terminus of the two proteins is removed.

We also generated cDNAs that encode the 90 amino acids BEN domain and the remaining C-terminal amino acids. As shown in *Figure 6A*, we failed to detect a shift with the GST-heterodimers between these two 'BEN domain' proteins (lane 17). It is possible that the close proximity of the GST moiety in these proteins precludes formation of a dimer that can form a stable DNA–protein complex. Alternatively, sequences in the extended 130 amino acid homology region might be important in facilitating DNA binding (e.g., aligning the two BEN domains).

As for the N- and C-terminal halves of Elba3, neither supported complex formation when mixed with the two other Elba proteins (not shown) nor did they supershift the shift generated by the GST-Elba1F:GST-Elba2F dimer (see *Figure 6E*).

## Expression of *elba1* and *elba3* is developmentally restricted

One important question is the basis for the developmentally restricted activity of the Elba factor. As all three proteins co-migrate with yolk protein they are difficult to visualize or quantitate by Western blots. For this reason we used Northerns to examine the expression of the three *elba* genes during development. These experiments indicate that the DNA binding/boundary activity of the Elba factor is developmentally restricted because of the temporal expression patterns of the paired *elba1* and *elba3* genes. *Figure 7* and *Figure 7—figure supplement 1* show that neither is expressed during oogenesis, and that their mRNAs are largely absent in 0–2 hr embryos. Consistent with the previous identification of *elba1* as a mid-blastula transition gene (*Singer and Lengyel, 1997*), expression of *elba1* and also *elba3* peaks at the blastoderm stage (2–4 hr), and then disappears. The disappearance of the *elba1* and *elba3* mRNAs occurs in the same time frame as the loss of boundary activity in transgene embryos and DNA binding activity in nuclear extracts. In much older embryos *elba3* mRNAs are detected, but are present at a lower level than in the blastoderm stage (see also *Figure 7—figure supplement 1*). While expression of *elba1* and *elba3* is developmentally restricted, *elba2* and its neighboring ortholog, *insv*, appear to be expressed throughout much embryogenesis. Moreover, both are expressed during oogenesis and likely are maternally deposited, as high levels are present in 0–2 hr embryo before zygotic transcription is activated.

## Elba factor is bound to *Fab-7* in during early but not mid-embryogenesis

Taken together, the findings described above suggest a plausible model for why the insulator activity of boundaries depending on the Elba recognition sequence is restricted to early embryos. A functional hetero-tripartite Elba factor would be assembled in early embryos and would bind to boundary elements containing the Elba recognition sequence (*Figure 8C*). Later in development, after the *elba1* and *elba3* mRNAs disappear, the hetero-tripartite Elba factor would disappear as well. In this case, Elba2 wouldn't be able to bind to boundaries containing the Elba recognition sequence on its own. We used chromatin immunoprecipitation (ChIP) of staged 2–5 hr and 9–12 hr embryos to test

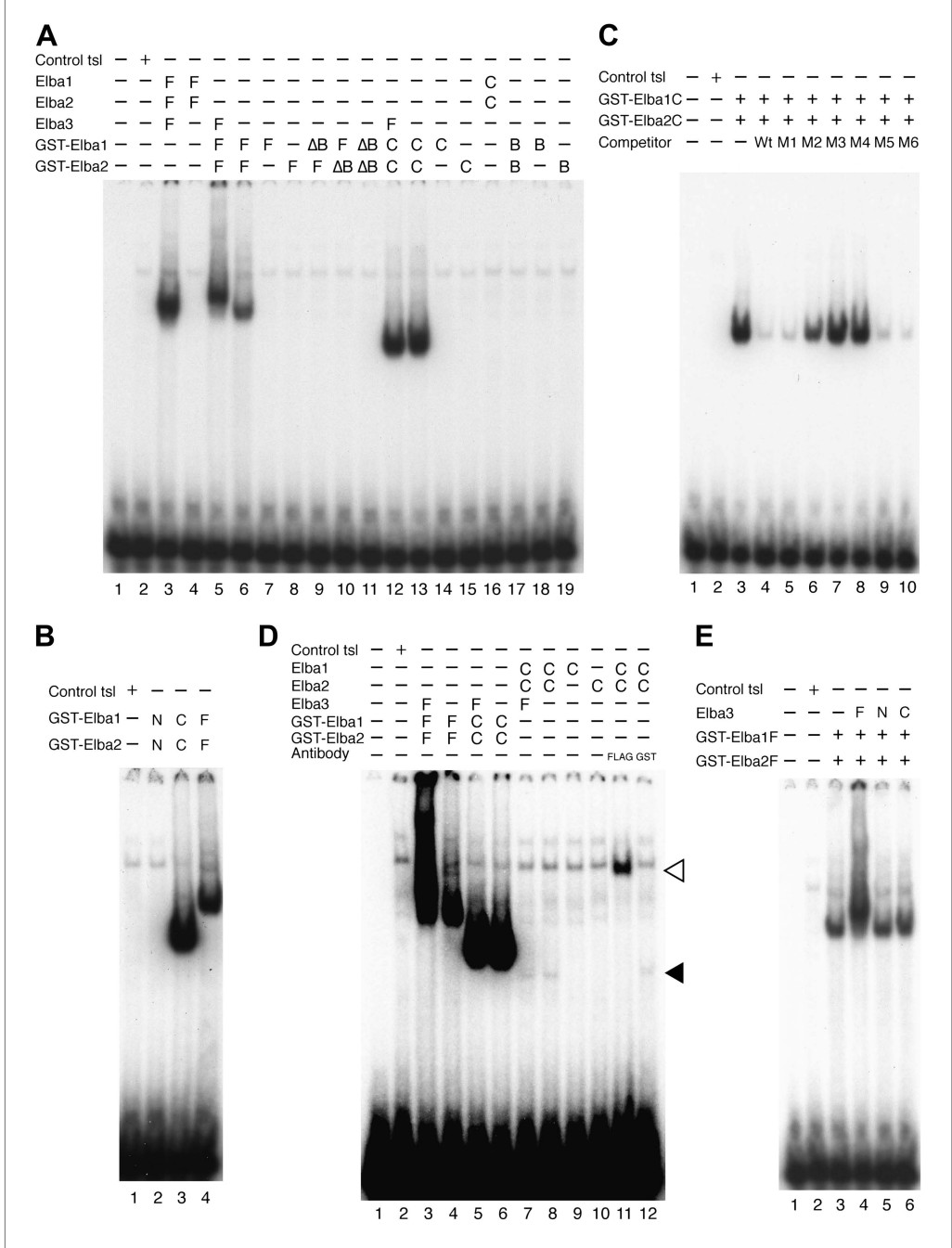

**Figure 6**. Characterization of Elba complex. (**A**) DNA binding activity of different Elba protein variants. The proteins were translated in vitro from the mixed mRNAs shown above the lanes. All proteins used in this figure are FLAG tagged and approximately the same amounts of the translated proteins were added to each lane. The identities of the proteins added to each lane are indicated above the lane. For example, lane 3 has all three full length Elba proteins, while in lane 5 full length Elba3 is combined with full length Elba1 and Elba2 proteins that have an N-terminal GST tag. Lane 13 has GST fused to the C-terminal half of Elba1 and Elba2, while in lane 16 the C terminal halves of Elba1 and Elba2 lack the GST moiety. (**B**) GST-fused Elba1F:Elba2F (F) or Elba1C:Elba2C (C) binds to the Elba probe while Elba1N:Elba2N (N) does not. (**C**) Sequence specificity of the GST-Elba1C:GST-Elba2C dimer is the same as the nuclear/reconstituted hetero-tripartite complex (**Figure 4**). The EMSA experiment with GST-Elba1C:GST-Elba2C were performed in the absence (lanes 1–3) or presence of 100-fold excess of non-labeled competitors as indicated above the lanes. The sequences of competitor DNAs are shown in **Figure 1**. (**D**) Elba1C and Elba2C proteins lacking the N-terminal GST moiety have a weak DNA binding activity. Position of

*Figure 6. Continued on next page*

*Figure 6. Continued*

Elba1C:Elba2C shift is indicated by closed arrowhead. Position of the Elba1C:Elba2C FLAG supershift is indicated by open arrowhead. Proteins added to each lane including FLAG and GST antibodies are indicated above the lane. Note that the same shift was detected in lane 16 of (**A**) when the X-ray film was exposed for a longer period of time. (**E**) Supershifts of the GST-Elba1F:GST-Elba2F generated by the addition of Elba3. Proteins corresponding to the full length Elba3 (F), or the N-terminal (N) and C-terminal (C) halves of Elba3 were added to an EMSA reaction mix containing the GST-Elba1:GST-Elba2 dimer. Proposed structures of the native Elba complex, the artificial GST-dependent dimer by GST-Elba1:GST-Elba2 and GST-Elba1C:GST-Elba2C are shown in *Figure 6—figure supplement 1–3*, respectively.

The following figure supplements are available for figure 6.

**Figure supplement 1**. A schematic model for the tripartite Elba complex.

**Figure supplement 2**. The artificial GST-Elba1:GST-Elba2 heterodimer binds to the asymmetric Elba recognition sequence.

**Figure supplement 3**. The artificial heterodimer of GST-Elba1C:GST-Elba2C binds to the asymmetric Elba recognition sequence.

this model. As standard formaldehyde based ChIP protocols gave inconsistent enrichment of *Fab-7* sequences, we adapted the cross-linking procedure of *Nowak et al. (2005)* for Drosophila embryos. *Figure 8A,B* shows that as predicted Elba1 and Elba2 are bound to the *Fab-7* Elba sequence in 2–5 hr embryos, but not to control sequences from the *twine* (*twe*) and *Sex-lethal* (*Sxl*) genes. In contrast, neither protein appears to be bound to *Fab-7* in 9–12 hr embryos. These findings dovetail nicely with the developmental profiles of Elba blocking activity in transgene assays and Elba binding activity in nuclear extracts. In addition, even though *elba2* is expressed in 9–12 hr embryos, the Elba2 protein isn't found associated with the *Fab-7* Elba sequence in the absence of the two other Elba proteins.

## All three Elba proteins are required for early boundary activity

We sought evidence connecting Elba binding in vitro and in vivo to *Fab-7* and its boundary activity during early embryogenesis. Previous studies indicate that a combination of functionally redundant factors, not just Elba, generate *Fab-7* boundary activity during early embryogenesis and at other stages of development (*Mihaly et al., 1997*). Consequently, the activity of the intact *Fab-7* boundary in the context of BX-C or in transgene assays is not expected to be disrupted by knocking down the Elba factor or even by mutations in the Elba binding site. On the other hand, unlike the intact boundary, Elba seems to play a more critical role in the insulating activity of the pHS1 multimer (*Aoki et al. 2008*). For this reason, we injected double-stranded RNA (dsRNA) specific for each Elba protein into embryos transgenic for the pHS1×4 pCfhl boundary reporter. As a control for specificity, we also injected dsRNA specific for the BEN domain gene *insv*, the close relative of *elba1* and *elba2*.

pChfl has two *ftz* enhancers and a LacZ reporter. The UPS enhancer drives LacZ expression in a stripe pattern during early embryogenesis, while the NE enhancer drives expression in the CNS during mid-embryogenesis. *Figure 9A* shows that both enhancers activate expression at the appropriate stage when a random (or DNA) is interposed between them and the reporter. When pHS1×4 is placed between the enhancers and the reporter, it blocks the UPS stripe enhancer from activating LacZ expression in early embryos, but unlike the full length *Fab-7* boundary (see *Figure 1—figure supplement 1*) it doesn't block the NE enhancer in mid-embryogenesis. Note that like *Fab-7*, pHS1×4 must be interposed between the enhancers and the reporter to block activation (*Schweinsberg and Schedl, 2004*).

Buffer injection had no apparent effect on pHS1×4 blocking of the UPS stripe enhancer. As observed for uninjected embryos, more than 50% of the buffer injected controls in each experiment were Class 1 and did not express LacZ (*Figure 9B*). In contrast, UPS blocking activity was disrupted when embryos were injected with dsRNA specific for one of the Elba proteins. For both *elba1* and *elba3*, most fell into Class 3–5, while for *elba2* most were Class 2–4. The loss of boundary activity is specific to constituents

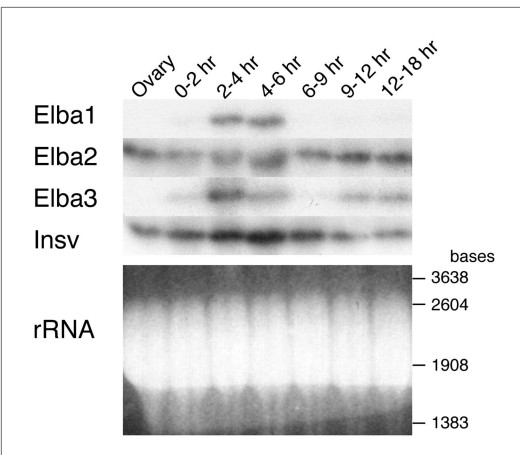

**Figure 7**. Expression of mRNAs encoding Elba factors and Insv. Total RNA from ovaries and staged embryos as indicated were probed with cDNAs encoding each protein. Shown on the bottom is the ethidium bromide staining of a gel for the Northern blotting as a loading control. The positions of the RNA markers are indicated on the right.
The following figure supplements are available for figure 7.

**Figure supplement 1**. Loci of Elba proteins and the expression patterns of their mRNAs during the development.

of the Elba complex, as injection of dsRNA for the closely related *insv* did not disrupt boundary activity. Similar results were obtained in two other sets of pHS1×4 dsRNA injection experiments, and in experiments using embryos transgenic for the Elba multimer (*Figure 9—figure supplement 1*). Like the set shown here, the disruption of boundary activity in these experiments was greatest for *elba1* and *elba3*, while smaller effects were observed for *elba2*. The differences in expression pattern of the *elba* mRNAs most likely accounts for this differences in sensitivity to dsRNA. While expression of *elba1* and *elba3* would be turned on in most instances after injection of the dsRNA, there is a substantial pool of maternally derived *elba2* mRNAs that could already be translationally engaged at the time of injection. Likewise the variability in the loss of boundary activity in *elba* injections most probably reflects differences in age of the embryos when they were injected, as well as variations in the amount and location of the injected dsRNAs.

## Discussion

The constitutive insulator activity of the *Fab-7* boundary is generated by a series of functionally redundant sub-elements whose activity is developmentally restricted. Here we report the identification and characterization of a factor, Elba, which confers the insulator activity of one of these sub-elements in early embryos. Elba is an unusual hetero-tripartite complex. It consists of two ~40 kDa proteins, Elba1 and Elba2, that have a C-terminal 90 amino acid BEN domain embedded in a larger 130 amino acid region of homology. The BEN domain is found in insect and vertebrate nuclear proteins and has been implicated in protein:protein interactions and transcriptional regulation (*Abhiman et al., 2008*). The third protein, Elba3 has no distinctive domains and seems to be limited to the Drosophilids. All three proteins are present in the Elba complex detected in nuclear extracts and are required to reconstitute Elba DNA binding in vitro.

Our functional studies indicate that DNA binding is mediated by the C-terminal domains of Elba1 and Elba2; however, in order to form a DNA binding 'pocket' that recognizes the asymmetric Elba sequence, CCAATAAG, Elba1 and Elba2 must be linked in a 'heterodimer' (*Figure 5C*). In the tripartite Elba complex, this seems to be the role of Elba3. It brings Elba1 and Elba2 together by interacting with sequences in the N-terminal half of the two proteins. These conclusions are supported by a number of findings. First, it is possible to circumvent the requirement for Elba3 by fusing a heterologous dimerization domain, in this case GST, to the N-terminus of full length Elba1 and Elba2. When co-translated, these two fusion proteins shift the Elba probe without Elba3. On the other hand, they can still interact with Elba3 as the shift generated by the two proteins can be supershifted by the addition of Elba3. Second, the Elba1C and Elba2C proteins, which span the 130 amino acid C-terminal homology region, can reconstitute full DNA binding activity when fused to GST. Since GST is known to dimerize, but not form higher order complexes, this would argue that the Elba DNA binding pocket is likely formed by a GST-Elba1C:GST-Elba2C dimer rather than by some other more complicated combination of the two proteins. Moreover, this binary complex has the same sequence specificity as the native heterotripartite complex. On the other hand, the GST-Elba1C:GST-Elba2C proteins differ from the full length GST fusions in that they don't interact with Elba3.

While these findings are consistent with the model for the Elba complex shown in *Figure 5C*, there are several unresolved but important issues. For one, though our experiments indicate that the BEN domain is essential for DNA binding, we were unable to demonstrate that it is sufficient. It is possible

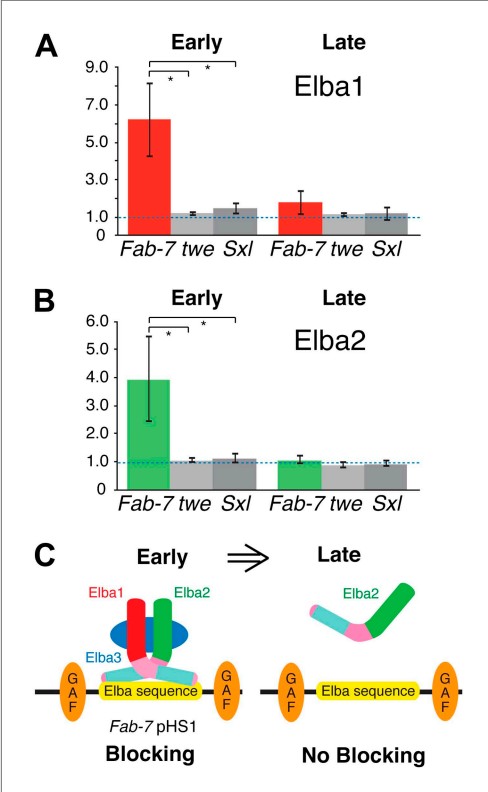

**Figure 8**. Elba proteins are bound to *Fab-7* in early but not late embryos. (**A**) and (**B**) Elba1 and 2 ChIPs. Early (2–5 hr) or late (9–12 hr) embryos were cross-linked and after processing immunoprecipitated with Elba1 or Elba2 antibodies, or pre-immune serum. Sequences from the pHS1 region of *Fab-7* or control *twine* and *Sex-lethal* sequences were detected by qPCR. The y-axis shows the average immune/preimmune ratio. *p<0.05. (**C**) Model showing binding and enhancer blocking by the Elba complex in early but not late embryos. GAF: GAGA factor and its binding sites.

that steric hindrance from the closely linked GST moiety prevents formation of the DNA binding pocket; alternatively, the extended homology region may contain elements that are important for DNA binding. Another apparent anomaly is that the homodimers formed by the full length or the C-terminal GST fusions don't appear to bind to the Elba probe. Since we've found that the Elba factor can bind to variants of the Elba sequence, albeit with reduced affinity, we would have expected that the homodimers would exhibit at least some evidence of DNA binding activity. Further studies will be required to resolve these issues.

As would be the case for many other bounda-ries, the presence of functionally redundant elements in the full length *Fab-7* precludes a direct demonstration that a single factor like Elba is needed for *Fab-7* insulating activity either in the context of BX-C or in transgenes assays. However, several lines of evidence argue that Elba does in fact have such a function for the endogenous *Fab-7* boundary. To begin with, previous studies showed that the insulating activity of the 236-bp pHS1 *Fab-7* sub-element in early embryos is com-promised when the Elba recognition sequence is mutated. Moreover, when multimerized, the Elba sequence is sufficient on its own to confer insulat-ing activity. That the Elba factor is responsible for this insulating activity is supported by the effects of RNAi knockdowns in embryos carrying either the pHS1×4 transgene or the Elba sequence mul-timer. For all three Elba proteins, RNAi knock-downs compromises the insulating activity of both pHS1×4 and the Elba multimer. In contrast, knockdowns of the closely related BEN domain protein, Insv, have no effect.

Our findings would also explain why the insu-lating activity of the Elba factor/Elba sequence is developmentally restricted. Two of the three Elba proteins, Elba1 and Elba3, are encoded by genes that are active during the mid-blastula transition, but not later in development. Since the hetero-tripartite complex is required to reconstitute DNA binding activity, Elba insulating activity would be expected to peak during the blastoderm/early gastrula stage when high levels of *elba1* and *elba3* mRNA are expressed. However, it should gradually dissipate after transcription of *elba1* and *elba3* ceases. While we don't know precisely when Elba1 and Elba3 disappear, there is little Elba DNA binding activity in nuclear extracts of 6–12 hr embryos. Moreover, even though the *elba2* gene is expressed throughout development, it is not found associated with *Fab-7* in 9–12 hr embryos. Importantly, the results of the EMSA and ChIP experiments provide strong support for the idea that the requirement for Elba activity evident in the RNAi knockdowns reflects a direct role in insulating activity rather than indirect role.

One important question is why does a constitutive insulator like *Fab-7* utilized developmentally limited factors like Elba? A plausible explanation comes from the finding that boundaries are not autonomous entities, but rather function in combination (likely pairwise) with other boundaries (*Cai and Shen, 2001*; *Muravyova et al., 2001*). Both boundary competition (*Gohl et al., 2011*) and bound-ary bypass experiments (*Kyrchanova et al., 2008a*; *Maksimenko et al., 2008*) indicate that some combinations are functional while others are not. Moreover, how different boundary combinations work together depends upon developmental stage and tissue (*Gohl et al., 2011*). Thus, the use of

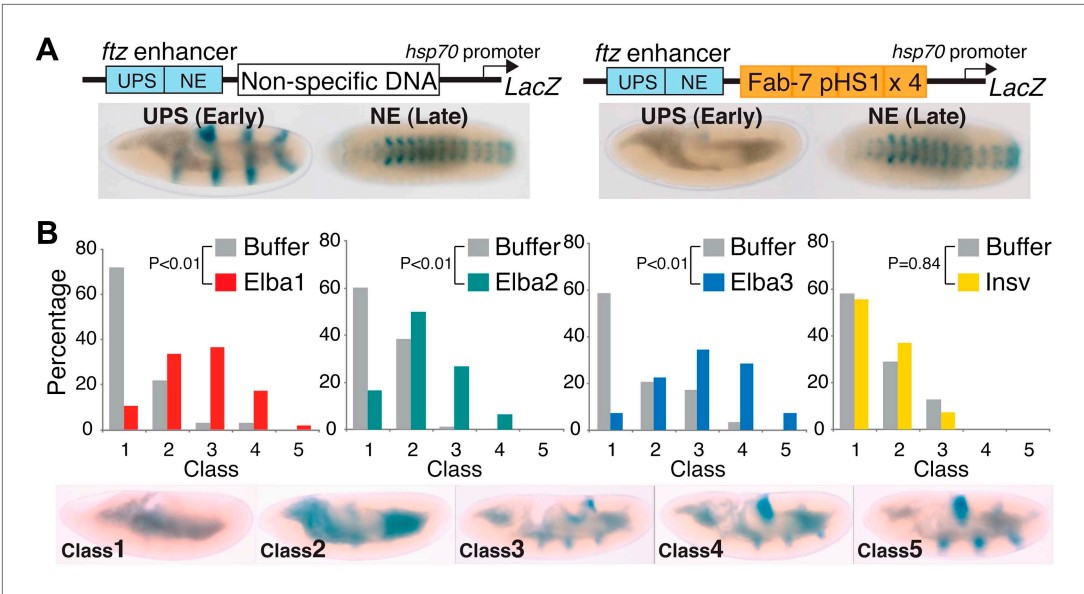

**Figure 9**. Hetero-tripartite Elba complex mediates early boundary activity. (**A**) Top: *fushi tarazu* (*ftz*) enhancers drives stripe expression (UPS) in stage 10–11 embryos and central nervous system expression (NE) in stage 13–14 embryos. Bottom: Four copies of *Fab-7* pHS1 (which contains the Elba sequence) blocks the UPS, but not the NE enhancer. *elba1-3* and *insv* dsRNAs or buffer alone were injected into embryos transgenic for the *ftz*-4×pHS1-LacZ enhancer blocking reporter and blocking activity examined. In each experiment ~100 embryos were photographed and categorized into Class 1–5 as indicated. Graphs show the percentage of embryos in each class for RNAi and the buffer control in that experiment. Three independent injection experiments were done for each protein. All gave similar results and only one is shown here. p values from the t-tests are shown in the graphs.

The following figure supplements are available for figure 9.

**Figure supplement 1**. Double-stranded RNA injection into embryos transgenic for the Elba×8 multimer reporter and a reporter containing non-specific DNA in the blocking position.

stage specific factors at *Fab-7* could reflect a need to optimize combinations with other nearby BX-C boundaries or the Abdominal-B promoter (***Kyrchanova et al., 2008b***, ***2011***). A number of findings are consistent with this idea. When heterologous boundaries are used to replace *Fab-7* in BX-C, they are unable to provide bypass activity, while their insulating activity can be lost in a stage and tissue specific fashion (***Hogga et al., 2001***). In contrast, *Fab-7* is able to replace *Fab-8* (***Iampietro et al., 2008***). Likewise, the boundary activity of *Fab-7* in a foreign environment is not only context dependent, but also varies during development (***Gohl et al., 2011***). Probably the most direct evidence that boundaries in BX-C like *Fab-7* are closely matched with their flanking neighbors comes from the boundary bypass experiments of ***Kyrchanova et al. (2011)***. They found that bypass is observed when *Fab-7* is combined with the neighboring boundaries *Fab-6* and *Fab-8*, while bypass interactions with boundaries from elsewhere in BX-C are only weak at best. Interestingly, in the cases that have been analyzed most thoroughly, boundary bypass is mediated by homologous protein:protein interactions (***Kyrchanova et al., 2008a***). Thus, bypass is observed when multimerized binding sites for the dCTCF boundary factor are paired, but not when dCTCF binding sites are paired with sites for example Zw5. Though *Fab-8* differs from *Fab-7* in that it utilizes dCTCF (***Holohan et al., 2007***; ***Kyrchanova et al., 2011***), EMSA experiments with 0–6 hr and 6–12 hr nuclear extracts indicate that they also have binding sites for several of the same stage specific factors (***Aoki et al., 2008***; Wolle et al., unpublished data). These include sites for the Elba factor, and sites for two different late (6–12 hr) stage specific factors. Conceivably this could also be true for *Fab-6*.

Another question is the fate of Elba2 after Elba1 and Elba3 disappear. While Elba2 can't bind to the Elba sequence on its own, it could function as an insulator in other contexts using different protein partners and presumably also recognition sequences. A plausible partner would be the *Notch* signaling

pathway protein Insv, which is encoded by a tightly linked gene that has a similar expression pattern to *elba2*. Insv and its closest mammalian relatives, the NAC proteins (which are thought to function as stem cell pluripotency genes, as well as in cocaine addiction and cancer), are believed to regulate transcription by acting as co-repressors (*Cha et al., 1997*; *Mackler et al., 2000*; *Wang et al., 2006*; *Korutla et al., 2009*; *Duan et al., 2011*). However, since our findings indicate that the BEN domains of Elba1 and Elba2 are essential for DNA binding, it would be reasonable to suppose that Insv as well as the mammalian counterparts are not co-repressors, but are rather sequence specific DNA binding proteins. Interestingly, like members of the BEN domain protein family, CTCF was initially thought to be a transcriptional repressor (*Filippova et al., 1996*). Thus, a seemingly plausible speculation would be that other members of the BEN family besides the two Elba proteins function as insulators restricting the action of nearby enhancer elements instead of directly repressing transcription. Potentially supporting this idea is the finding that one of the mammalian BEN domain proteins, SMAR1, is a component of the nuclear matrix and associates with Matrix Attachment Regions (MARS) (*Chattopadhyay et al., 2000*). If this were the case, this would mean that CTCF is not the only vertebrate insulator protein. Also as is case for the Elba complex, modulating the expression or activity of Elba2, Insv or NAC1 in different tissues or cell types could change the regulatory domain landscapes, and potentially alter global patterns of gene expression.

## Materials and methods

### Embryo collection and storage

Oregon R population cages were used to collect embryos of the appropriate stage on apple juice plates. The embryos were washed off the plates, dechorionated with 50% bleach (2.6% sodium hypochloride) for 3 min, rinsed with 0.12 M NaCl/0.04% Triton X-100 and then 0.12 M NaCl, and frozen in liquid nitrogen. Embryos were stored at −80°C until the extraction. The Canton S embryos (generous gifts from Dr. James T Kadonaga and Dr. Rock Pulak) of various ages were also used for testing different steps in the purification procedure.

### Preparation of embryonic nuclear extracts

Small-scale embryonic nuclear extracts were prepared from 10 g of 0- to 6-hr-old Oregon R embryos as described previously with small modifications (*Aoki et al., 2008*). The frozen embryos were suspended in 30 ml homogenization buffer (HB; 3.75 mM Tris–HCl pH7.4/0.5 mM EDTA–KOH pH7.4/20 mM KCl/0.05 mM spermine/0.125 mM spermidine/0.5% 2,2′-thiodiethanol/2 µg/ml aprotinin/0.1 mM phenylmethylsulfonyl fluoride: PMSF/0.1% digitonin) and disrupted using a motorized teflon-glass homogenizer for 10 strokes and then with a glass–glass Potter homogenizer for an additional 10 strokes. The lysate was filtered through two layers of Miracloth (EMD Millipore, Billerica, MA) to remove debris and centrifuged at 1900×*g* for 5 min in a swinging bucket rotor. The resulting nuclear pellet was resuspended and washed four times with 40 ml of HB and finally suspended in 2 ml of Nuclear Extraction Buffer 20 (NEB20: 10 mM HEPES–KOH pH7.4/20 mM KCl/3 mM $MgCl_2$ 0.1 mM EDTA 10% glycerol/1 mM dithiothreitol: DTT/0.2 mM PMSF/2 µg/ml Aprotinin). The nuclear suspension was transferred to a polyallomer ultracentrifuge tube (Sarstedt 65-90219) and an equal volume of NEB 700 (same as NEB 20 except that the KCl concentration was 700 mM) was added to the suspension to give a final concentration of 360 mM KCl. After 30 min incubation at 4°C, the sample was centrifuged at 150,000×*g* for 1 hr at 4°C in a Beckman SW50.1 swinging bucket rotor. The resulting supernatant (~ 4 ml) was divided into aliquots for storage at −80°C.

For the purification of the Elba factor nuclear extracts were prepared from 60 g (first procedure) or 100 g (second and third procedures) of Oregon R 0–12 hr (first and second) or 0–6 hr (third) embryos. The embryos were divided into aliquots of 10 g each and the extract was prepared as described above except that 1.0% 'Nonidet P-40 substitute' (Sigma-Aldrich, currently designated as 'Igepal CA-630') was used instead of 0.1% digitonin in HB. Because Elba activity is destabilized by freezing–thawing and dilution, the extracts were processed through the phenyl-TSK hydrophobic chromatography (see 'Pre-purification') on the same day as the extraction.

### Electrophoresis mobility shift assays (EMSAs)

The Elba probe (*Figure 1D*) was labeled with $^{32}$P and used for EMSA assay under the same conditions as described previously (*Aoki et al., 2008*) except for the concentration of the non-specific competitor poly (dI-dC):poly(dI-dC) in the binding reaction. The final concentration of poly(dI-dC):poly(dI-dC) was

varied between 12.5 and 250 µg/ml depending on the relative purity of Elba factor. The salt concentrations in the samples were adjusted so that the final concentration of KCl would be about 0.1 M in the binding reaction.

For the EMSA experiments shown in *Figure 4*: (A) Control 'nuclear extracts' correspond to 1 µl of 0–6 hr nuclear extracts (corresponding to about 2.5 mg of embryos) from the small scale preparation described above. (B) In the competition experiments shown in *Figures 4C* and *6C*, the indicated cold competitor DNA was present in 100-fold excess over the labeled probe. (C) For the EMSA experiments using the three in vitro-translated proteins: In *Figure 4A*, 6 µl of the in vitro translated proteins were used in each lane. In *Figure 4B,C*, 1 µl of each in vitro translated protein was used by itself or in combination as indicated. The total amount of rabbit reticulocyte lysate was adjusted between the lanes so that the non-specific DNA-binding activities would be the same. (D) In the 'super-shift' experiment in *Figure 4D*, 1 µl each of pre-immune or immune serum was added to the incubation mix containing 1 µl of the small-scale nuclear extract.

## Isolation of the Elba factor

### Pre-purification

*Figure 2* shows a schematic of the Elba purification procedure. The nuclear extract was processed in four sequential column-chromatography steps using the loading and elution conditions indicated in the diagram. All chromatographic steps were performed at 4°C by gravity flow using Econo Columns (Bio-Rad). The distribution of Elba in the fractions was monitored by EMSA using the $^{32}$P-labeled Elba probe (e.g., *Figure 3A*). In general, HEPES-based buffers (HBB: 10 mM HEPES pH 7.4/0.1 mM EDTA–KOH/10% glycerol) supplemented with different concentrations of salts were used for equilibration, washes, elution and dialysis as indicated in *Figure 2*. After the Phenyl-TSK chromatography step, 0.05% of Nonidet P-40 and 1 mM of dithiothreitol (DTT) was added to all buffers. In the wash/elution steps in the chromatography, the salt concentrations were changed in a stepwise fashion. Because we found the Elba activity was rapidly lost upon dilution or dialysis, we kept Elba concentrated and the dialyses were limited to 2 × 1.5 hr.

In order to maximize the amount of Elba factor recovered for affinity purification, the Elba activity in the flow-through of the S-Sepharose chromatography was recovered and subjected to the pre-purification steps above by using reduced amounts of beads or matrix until the flow through from the S-Sepharose column no longer contained significant amounts of Elba activity. All of the 0.5 M KCl S-Sepharose fractions containing Elba activity were combined and subjected to stepwise dialyses first against HBB/0.3 M KCl/0.05% Nonidet P-40/1 mM DTT for 1.5 hr and then against HBB/0.15 M KCl/0.05% Nonidet P-40/1 mM DTT for additional 1.5 hr. Before the affinity purification, a non-specific DNA competitor poly(dI-dC):poly(dI-dC) was added to the samples in the final concentration of 0.015 µg/µl.

### Preparation of DNA affinity beads

Oligonucleotides corresponding to the top and bottom strands of three tandem wild-type (3× Elba-site) or mutated Elba sites (3× Elba-M3-site: the mutation M3 is shown in *Figure 1D*) were synthesized and purified by PAGE (*Supplementary file 1*). The bottom strand of each DNA was synthesized with a biotin moiety and two overhanging bases at the 5' end. 10 nmol of each strand were mixed in a final volume of 1 ml TE (10 mM Tris–Cl pH7.5/1 mM EDTA) + 100 mM NaCl and annealed. The double strand wild type and mutant DNA was then incubated with 200 µl of streptavidin-conjugated agarose beads (Pierce) overnight at 4°C followed by the centrifugation to remove free DNA. More than 90% of the double-strand DNA appeared to be coupled with the streptavidin–agarose beads (>45 pmol DNA per µl beads, or >135 pmol binding sites per µl beads). The beads were washed three times with TE + 0.1 M NaCl and then stored at 4°C in TE/0.1 M NaCl/0.1% sodium azide.

### DNA affinity purification of Elba
#### Single affinity purification experiments

The pooled and dialyzed S-Sepharose Elba fractions (3 ml first experiment or 6 ml second experiment) were divided into two equal parts and incubated in ~1 ml aliquots with 3× Elba-site or 3× Elba-M3-site (50 µl first or 80 µl second) affinity beads. After incubating an hour, the microfuge tubes were spun at 1000×*g* for 1 min, the supernatants removed and a new aliquot added. Then the beads were washed and eluted with increasing concentrations of KCl as indicated in *Figure 2*. For each salt elution, the

beads were washed three times with one bead volume of buffer and the supernatant from each salt elution was combined in a single tube. EMSA experiments showed that bulk Elba activity from the mutant affinity beads was in the unbound fraction, while most of the Elba activity from wild type beads was in the 1.0 M KCl fraction. As described in the text the 1.0 M KCl fraction from the wild type beads (designated WE1) and the mutant beads (ME1) were subjected to mass-spectrometry.

## Cross affinity purification experiments

Several problems were evident in the single affinity purification experiments. First, there were too many candidate proteins making identification of the Elba factor impractical. Second, the large number of proteins in ME1 suggested that there was significant amount of non-specific binding to the affinity beads. Third, the ME1 sample contained a small amount of Elba activity raising the possibility that the Elba factor was included in the list of proteins detected by mass spectrometry in both the WE1 and ME1 samples. To address these problems, we needed a procedure that would substantially reduce the number of non-specific proteins and at the same time give much greater differences in Elba yield from wild type and mutant affinity beads. For this purpose we used a cross affinity purification scheme (*Figure 2*). The sample from S-Sepharose fractionation was divided into two parts (about 6 ml each) and incubated with wild type or mutant DNA affinity beads (60 µl each) as described above. The unbound supernatant from the wild type (WS) and the mutant (MS) affinity beads was recovered and used for cross affinity purification. The WS fraction was incubated with mutant DNA affinity beads (33 µl) while the MS fraction was incubated with wild type DNA affinity beads. The beads were washed with buffer containing increasing amounts of KCl and then eluted with 1.0 M KCl. The difference in Elba activity between wild type (MSW1) and mutant (WSM1) 1.0 M KCl fractions became much greater for the cross-affinity purified samples than it was for the corresponding 1.0 M fractions from the single step samples (WE1 and ME1). Additionally, there was also a substantial reduction in the total number of proteins in the MSW1 and WSM1 fractions compared to the single affinity WE1 and ME1 fractions from the first and second experiments.

## Protein precipitation, digestion with protease and mass-spectrometry

Trichloroacetic acid (TCA; Sigma-Aldrich) was added to the 1.0 M KCl fractions from the affinity beads so that the final concentration of TCA was 25%. After an overnight incubation at 4°C, the proteins were collected by centrifugation at 20,000×*g* for 30 min. The protein pellets were washed two times with ice-cold 30% TCA and then two times with ice-cold acetone. The dried protein samples were subjected to trypsin digestion followed by electrospray ionization mass-spectrometry (ESI-MS) as described previously (*Washburn et al., 2001*; *Bern et al., 2004*). The software DTASelect and Contrast were used for peptide data analyses and protein predictions (*Tabb et al., 2002*).

## In vitro transcription-translation of Elba factors

Approximately 5 µg of plasmids encoding Elba cDNA clones were digested with appropriate restriction enzymes and used as templates for in vitro transcription. The capped mRNAs were transcribed using T3 RNA polymerase (for Elba2; Promega) or T7 RNA polymerase (for Elba1 and Elba3; USB) in the presence of ribonucleotide-triphosphates (NTPs) and the cap-analog m7GpppG (Promega) at 40°C for 1 hr. After removing the plasmids with RNase-free DNase (RQ DNase; Promega), the mRNAs were extracted with phenol/chloroform and precipitated with ethanol. The mRNA pellets were suspended in 50 µl of RNase-free water and 3.5 µl each was used for one translation reaction of 25 µl. (In the mixed translation of three proteins in *Figure 4A*, 1.17 µl of each mRNA was used.) A rabbit retiulocyte lysate (Promega) was used for in vitro-translation reactions. To confirm that the mRNA directs the synthesis of a protein of the appropriate molecular weight, radio-labeled proteins were synthesized in parallel reactions containing [35]S-methionine/cystine (Tran[35]S-label; MP Biomedicals).

In the in vitro translations of tagged Elba mutant proteins in *Figures 5* and *6*, all the plasmids were digested with Asp718 (Roche Diagnostics) and transcribed with T7 RNA polymerase. The protein products were detected with anti-FLAG antibodies (M2 from Sigma-Aldrich or anti-DYKDDDDK 1E6 from Wako) in the Western blotting.

## Polyclonal antibody production

Rabbit polyclonal antibodies against the three Elba proteins were generated by injecting bacterially-expressed proteins into rabbits. The Elba proteins were expressed using the 6×His-T7 tag pET28

vector or the 6×His-HA tag from a modified pET15 vector. Because most of the bacterially expressed proteins from these pET vectors were insoluble, the recombinant proteins were isolated from the whole bacterial lysate by SDS-PAGE gels. The proteins were eluted from the PAGE gels with the Bio-Rad Electro-eluter 422. The recovered proteins were mixed with TiterMax Gold adjuvant (Sigma-Aldrich) and injected into two rabbits each. The 6×His-T7 tagged proteins (200–650 μg per rabbit) were used for initial immunization while the 6×His-HA tagged proteins were used for all of the boosting injections. For the boosting injections, 180–500 μg protein were injected every 4 weeks. 15–25 ml of blood were collected every 2 weeks, and the titers of the sera were checked by Western blotting. The reactive sera were stored at −80°C.

## Plasmids

cDNAs encoding the Elba factors were obtained from the Drosophila Genomics Resource Center (DGRC). The clone names: RE24665 (pFlc-1-CG12205/Elba1), LD10908 (pBluescriptSK(−)-CG9883/Elba2), LD42284 (pOT2-CG15634/Elba3). For in vitro transcription experiments, these plasmids were prepared using cesium chloride ultracentrifugation. The protein-coding region of CG3227/insensitive (insv) cDNA was obtained by RT-PCR using embryo RNA and subcloned into pBluescriptKS(+) (Stratagene).

For antibody production, the plasmids for 6×His-T7-tagged proteins were constructed by amplifying the protein-coding regions of Elba2 or Elba3 cDNAs with PCR and by introducing them into the Bam HI/Xho I sites of the pET28c vector (Novagen). The upstream PCR primers included a Bam HI site, whereas the downstream PCR primers had either Sal I or Xho I sites. In the case of the Elba1 cDNA, which has an internal Bam HI site, the full coding sequence was amplified using a 5′ PCR primer that had a Bam H1 site and 3′ primer that had a Sal I site. The PCR product was digested with Bam H1 and introduced into pET28c. The resulting plasmid was digested with Hind III (which cuts in the Elba1 cDNA) and Xho I (which cuts in the vector). It was then ligated to the Hind III-Sal I fragment generated by digestion of the full length Elba1 PCR product. The plasmids for 6×His-HA-tagged proteins were constructed using a two-step procedure. First, a modified pBluescriptKS(+) vector that contains the influenza HA tag at the Xba I–Bam HI sites (pKS(+)HA; Aoki, unpublished data) was used to attach the HA tag to 5′ end of each cDNA for the Elba proteins. The Elba cDNA fragments were generated by PCR and then introduced into at Bam HI/Sal I sites or Bam HI/Xho I sites of pKS(+)HA. The resulting HA-fused cDNAs were PCR amplified using an Xho I-HA-upstream primer and downstream T3 primer. The PCR products were digested with Xho I and introduced into the Xho I site of pET15b vector (Novagen).

A series of plasmids encoding protein tags were constructed based on the pBluescriptKS(+) (Stratagene) for the in vitro transcription of tagged Elba proteins in *Figures 5* and *6*. First a pKS(+) FLAG vector was generated by introducing a phosphorylated double-strand DNA fragment encoding a methionine followed by the FLAG tag at the Xba I and Bam HI sites. The resulting plasmid was cut at Bam HI and additional tags, such as GST, were introduced. For the FLAG-GST-fused proteins, pKS(+) FLAG-GST was constructed by introducing a PCR-amplified and Bgl II/Bam HI-digested cDNA fragment encoding GST into pKS(+)FLAG. The non-GST-tagged Elba proteins also had HA, c-myc or Streptavidin-binding peptide (*Keefe et al., 2001*) tags at their N-terminus. The partial cDNA fragments of Elba proteins were amplified by PCR and introduced into the protein tag-encoding vectors described above. Further details upon request.

## RNA extraction and Northern blotting

The total RNA samples were prepared from about 200–300 mg of appropriately aged embryos and dissected ovaries using the QuickPrep Total RNA Extraction Kit from Amersham-Pharmacia (#27-9271-01; currently discontinued by GE Healthcare). In the Northern blotting experiment 30 μg per lane of total RNA was applied to a 1.0% agarose gel (cast in 0.66 M formaldehyde/1× MOPS buffer [0.2 M 3-(N-morpholino) propanesulfonic acid/50 mM sodium acetate/10 mM EDTA]) and electrophoresis was performed in 1× MOPS buffer for 3 hr at 100 V. The separated RNA was transferred to a Zeta-Probe membrane (Bio-rad) by capillary blotting and cross-linked with a Stratalinker (Stratagene) UV cross-linker.

For probes, DNAs corresponding to the protein-coding sequences of Elba1-3 and Insv were prepared by PCR amplification or by digesting plasmids with appropriate restriction enzymes. They were $^{32}$P-labeled using α-$^{32}$P dCTP and Ready-To-Go DNA Labeling beads (-dCTP; GE Healthcare). After denaturating in boiling water, the probes were mixed with hybridization buffer (0.2 M sodium phosphate pH7.2/1% BSA/7% SDS) and hybridized with membranes overnight. The following day, the membranes were washed several times with 0.2× SSC/0.1% SDS at 65°C and exposed to the X-ray film.

## Chromatin immunoprecipitation (ChIP)

Staged 2–5 hr (early) and 9–12 hr (late) Oregon R embryos were collected, dechorionated and weighed. The embryos were cross-linked using a modification of the procedure of *Nowak et al. (2005)*. Chromatin IP steps were performed as previously described by *Kappes et al. (2011)* except that the washing of the immunoprecipitated beads was simplified to five successive washes with RIPA buffer only. One IP sample corresponds to approximately 100 mg of embryos. In each experiment a pair of immune-IP and pre-immune-IP samples were processed in parallel. 5 µl of immune or pre-immune serum was used for Elba1 (anti-Elba1 #1 in *Figure 4D*) and 2 5–10 µl each of serum was used for Elba2 (anti-Elba2 #1 in *Figure 4D*). The precipitated DNA samples were suspended in 30 µl of distilled water.

*Supplementary file 1* shows the primer pairs that were used to detect the target locus *Fab-7* HS1 (*Fab-7*) as well as two control loci *Sex-lethal* (*Sxl*) and *twine* (*twe*). The quantitative PCR (qPCR) was performed with triplicated 25 µl reactions using Power SYBR Green (Applied Biosystems) and 0.5 µl of sample DNA in Agilent M×3000P qPCR system. The ratio of immune/pre-immune precipitation was calculated by the comparative Ct method (ΔΔCT Method). The chromatin IPs were done four times for Elba1 and five times for Elba2 and the significance was determined using the unpaired t-test.

## RNAi injection

We used in vitro synthesized double strand RNAs for the RNAi injection experiments (*Kennerdell and Carthew, 1998*). A short 350- to 420-bp cDNA fragment was amplified from the *elba1-3* and *insv* cDNAs using complementary PCR primers that also contained T7 promoter sequences at their 5′ ends. The T7 promoter-cDNA fragments were purified from agarose gels and used as templates for T7 RNA polymerase (MEGA script T7; Ambion). The transcribed RNA strands were annealed by heating and then chilling, treated with DNase and purified by phenol/chloroform extraction and ethanol precipitation. The dsRNAs were suspended in injection buffer so that the final concentration was approximately 1 µg/µl (3.6–4.3 µM).

Transgenic embryos were collected on 10-cm apple juice agar plates from fly cups incubated at 18°C for 30 min. They were lined up on a piece of apple juice agar and then transferred onto double-stick tape attached to a cover glass. The embryos were covered with Halocarbon oil 27 and dsRNA was injected on the ventral side at the middle of the embryo. Each set of injected embryos was incubated on cover glasses at different temperatures (18°C, room temperature, or 25°C) so that they would develop to stage 10–11 at approximately the same time. The staged embryos were washed with PBS from the cover slips into a cup and then stained with X-gal as described previously (*Aoki et al., 2008*). For each experiment, buffer-injected embryos were also prepared and processed in parallel. Approximately 100 embryos from each dsRNA/buffer injection were photographed and the photographed embryos classified into Classes 1–5 according to the intensity of the *ftz* UPS enhancer driven LacZ stripe expression.

## Acknowledgements

We thank D Wolle and Drs F Karch, RK Maeda and R Mishra for communicating unpublished results and Drs JT Kadonaga and R Pulak for the generous gift of embryos. We also thank Dr G LeRoy for valuable advice on protein purification and chromatin IP.

---

## Additional information

### Funding

| Funder | Grant reference number | Author |
| --- | --- | --- |
| National Institutes of Health | GM043432 | Paul Schedl |
| National Institutes of Health | P41 RR011823 | John Yates |

The funder had no role in study design, data collection and interpretation, or the decision to submit the work for publication.

### Author contributions

TA, Conception and design, Acquisition of data, Analysis and interpretation of data, Drafting or revising the article; AS, Acquisition of data, Analysis and interpretation of data; JY, Acquisition of data, Analysis and interpretation of data; PS, Conception and design, Acquisition of data, Analysis and interpretation of data, Drafting or revising the article

# Additional files

## Supplementary files

• Supplementary file 1. The list of oligonucleotides used in this study. See also 'Materials and methods' for the use of each DNA.

## Major datasets

The following previously published datasets were generated

| Author(s) | Year | Dataset title | Dataset ID and/or URL | Database, license, and accessibility information |
|---|---|---|---|---|
| Gelbart WM, Emmert DB | 2010 | FlyBase High Throughput Expression Pattern Data Beta Version: | http://flybase.org/reports/FBgn0000227.html | Publicly available at Flybase http://flybase.org/ |
| Gelbart WM, Emmert DB | 2010 | FlyBase High Throughput Expression Pattern Data Beta Version: | http://flybase.org/reports/FBgn0031621.html | Publicly available at Flybase http://flybase.org/ |
| Gelbart WM, Emmert DB | 2010 | FlyBase High Throughput Expression Pattern Data Beta Version: | http://flybase.org/reports/FBgn0031434.html | Publicly available at Flybase http://flybase.org/ |
| Gelbart WM, Emmert DB | 2010 | FlyBase High Throughput Expression Pattern Data Beta Version: | http://flybase.org/reports/FBgn0031435.html | Publicly available at Flybase http://flybase.org/ |

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
