## [Decision Letter]

Thank you for choosing to send your work entitled "Elba, a novel developmentally regulated chromatin boundary factor has an unusual hetero-tripartite structure" for consideration at *eLife*. Your article has been evaluated by a Senior Editor and 3 reviewers, one of whom is a member of *eLife's* Board of Reviewing Editors. The following individual responsible for the peer review of your submission wishes to reveal his identity: Jim Kadonaga (Reviewing Editor).

The Reviewing Editor and the other reviewers discussed their comments before we reached this decision, and the Reviewing Editor has assembled the following comments based on the reviewers' reports.

This paper describes the purification of a sequence-specific DNA-binding protein termed Elba (Early Boundary activity), which binds to the pHS1 region in the *Fab-7* insulator element from the bithorax complex in *Drosophila*. Two of these polypeptides, Elba1 and Elba2, harbor a C-terminal BEN domain. The third polypeptide, Elba3, does not contain any conserved motifs. Beyond the biochemical scheme used to isolate the Elba complex, most of the data presented in the article are convincing gel shift experiments that bring insights into the mechanism of binding of the tripartite complex. In brief, Elba3 plays the role of an adaptor protein, bringing Elba1 and 2 together, thereby establishing a complex that recognizes the asymmetric sequence motif through the BEN domains of Elba1 and 2. It is interesting to note here that only the complex containing all three polypeptides can bind to the recognition site and that the different binary combinations of the subunits do not exhibit distinct binding to the Fab-7 target site. The genomic organization of the genes encoding Elba1-3 is intriguing. Indeed the *elba2* gene is found in the immediate vicinity of another BEN domain protein, Insensitive (Insv), involved in neurogenesis and Notch signaling. The *elba1* gene corresponds to a locus previously described by Judy Lengyel's laboratory as Bsg25A, based on its short expression window during embryonic development, at blastoderm stage (Bsg stands for Blastoderm-specific-gene).

Intriguingly, although *elba3* does not have sequence homology to *elba1*/Bsg25A, its transcription unit is located next to *elba1*. In addition to their close proximity, the *elba1* and *elba3* genes share the striking feature of being expressed during a very short and specific time window in early development at the blastoderm stage. This tight and specific regulation of *elba1* and *elba3* is in perfect agreement with the previous characterization of the Elba activity by Aoki et al. (2008) and represent a strong argument that the factors purified biochemically indeed correspond to the original Elba activity.

Ideally, a functional characterization with mutants would be the next best demonstration that the Elba protein complex mediates the early insulator activity of Fab-7. Unfortunately such mutants are not available and their generation is beyond the scope of this article. The authors nevertheless perform the difficult experiment of injecting embryos carrying their ftz-lac Z based insulator construct with double strand RNAs targeted at inactivating *elba1*, *elba2* or *elba3*. While the quality of the embryos that are depicted in Figure 10 is not optimal (probably reflecting the fact that they have to be injected and are thus limited in numbers), this last experiment nevertheless supports the idea that depleting *elba1*, *elba2,* or *elba3* interferes with the early enhancer-blocking activity of Fab-7.

The discovery that Elba is a new, BEN-domain dependent, heteromeric, sequence-specific DNA binding protein, and developmentally restricted boundary factor, is a significant finding. Boundaries represent an important class of regulatory elements and the identification of new boundary factors is of general interest. This work also establishes that the conserved BEN domain can contribute to DNA recognition in a sequence-specific manner.

Hence, the paper is appropriate for publication in *eLife* if it is revised in a manner that suitably addresses the specific comments below. In addition, the title should be modified because DNA-binding protein complexes with multiple polypeptides are not unusual.

1. Figure 10. In these assays, Elba could be functioning as a repressor rather than as a transcriptionally neutral boundary element. While either outcome would be interesting, it would be informative to address this issue. For example, the authors could use a reporter construct in which the "Fab-7 pHS1x4" fragment was upstream of the ftz enhancer.

2. Figure 10. It would be preferable if the authors tested two independent dsRNAs for the depletions as well as carried out some analysis of the extent of depletion (e.g., western blots, reverse transcription-PCR). Such experiments would strengthen this work. It is understood, however, that the parallel analysis of the three Elba subunits does provide some cross-validation.

3. The description of the isolation and identification of the Elba factor is long, too detailed and distracting. The authors should just state that the capacity of the initial affinity columns was exceeded and proceed with the sequential method that worked. 

4. Does the binding of Elba interfere with GAGA factor (GAF) binding?

5. A control that demonstrates the BEN-deletion mutant proteins are stable would be useful. One could argue these particular mutants do not bind the sequence because they are unstable.

6. Figure 8. How does the expression profile of the three Elba factors look at later stages of development? One could imagine that since the expression of Elba3 returns at later stages, the expression of Elba1 might increase post-18 hr. Thus, in later stage, the three Elba subunits could be present.

7. Have the authors tried to ChIP Fab-7 sequences with the Elba3 antibodies?

---

## [Author Response]

*1. Figure 10. In these assays, Elba could be functioning as a repressor rather than as a transcriptionally neutral boundary element. While either outcome would be interesting, it would be informative to address this issue. For example, the authors could use a reporter construct in which the "Fab-7 pHS1x4" fragment was upstream of the ftz enhancer*.

The requested experiment has been done. It was reported in Schweinsberg and Schedl (2004) and corresponds to the reporter with pHS1x4 in the NB (non-blocking position) in Figure 2 and Table 1. No silencing was observed for this 'upstream' construct, indicating that pHS1x4 must, like *Fab-7* itself, be placed between the enhancer and promoter to alter enhancer activity.

We also thought that this was a point that the readers might wonder about and for this reason we mentioned this old experiment parenthetically at the end of the section describing the dsRNA injections in Figure 10, which we have made more obvious to readers in the revised version.

*2. Figure 10. It would be preferable if the authors tested two independent dsRNAs for the depletions as well as carried out some analysis of the extent of depletion (e.g., western blots, reverse transcription-PCR). Such experiments would strengthen this work. It is understood, however, that the parallel analysis of the three Elba subunits does provide some cross-validation*.

Here the critical issue is whether the loss of blocking activity in our dsRNA injection experiments is due to depletion of Elba activity or arises from off-targets effects on some unknown factor(s). Typically concerns about off-target effects are addressed using a 2^nd^ dsRNA and showing that it has the same effect as the original. The supposition in this case is that if there are off-targets for the two dsRNAs, they will be different. We would argue that the finding that three independent dsRNAs, each targeting a different *elba* gene, disrupt boundary activity, while no effects are observed for the related gene *insv*, provides a much more rigorous demonstration of specificity, and more compelling evidence that the Elba factor is responsible for boundary activity, than would typically be provided in papers in which the authors use a second dsRNA to demonstrate that a specific gene/factor is required for a particular process. As for measuring the extent of depletion, this would be a very difficult experiment in the context of embryo injections and, importantly, would add little to the conclusions we draw from these experiments.

*3. The description of the isolation and identification of the Elba factor is long, too detailed and distracting. The authors should just state that the capacity of the initial affinity columns was exceeded and proceed with the sequential method that worked*.

Description of Elba isolation/identification has been shortened.

*4. Does the binding of Elba interfere with GAGA factor (GAF) binding*?

We don’t know the answer to this. While we know that the flanking GAGA sites are important for early boundary activity, we haven’t detected EMSA shifts that could be attributed to the GAGA binding sites either in early or late nuclear extracts. We have seen shifts for GAGA sequences elsewhere in *Fab-7* and *Fab-8* that we know, from supershift experiments, are generated by the GAGA factor. This would suggest that there is something odd about the two GAGA sites flanking the Elba sequence.

*5. A control that demonstrates the BEN-deletion mutant proteins are stable would be useful. One could argue these particular mutants do not bind the sequence because they are unstable*.

In our EMSA experiments we incubate the reaction mix for ½ hr at room temperature before loading it onto the gel. To address this question we incubated individual proteins in reaction mix for increasing amounts of time up to an hour and then analyzed stability by Westerns. We found that both the wild type and the deletion proteins underwent degradation and about three-quarters to one half was left after an hour. The mutant proteins also seemed to be slightly less stable. However, the small difference in stability would not account for our failure to detect DNA binding activity with the Ben deletions.

*6. Figure 8. How does the expression profile of the three Elba factors look at later stages of development? One could imagine that since the expression of Elba3 returns at later stages, the expression of Elba1 might increase post-18 hr. Thus, in later stage, the three Elba subunits could be present*.

Our Northerns and also ModEncode data suggest that *elba3* may be re-expressed in late embryos and during subsequent stages of development. By contrast, *elba1* doesn’t appear to be re-expressed (with the caveat that expression in only a single tissue or cell type might not have been detected). If this were correct, the Elba complex would not be re-formed. However, Elba3 could potentially link Elba2 to some other protein, e.g., Insv. It will be interest to explore this possibility.

*7. Have the authors tried to ChIP Fab-7 sequences with the Elba3 antibodies*?

We have not tried yet, but we are planning to do so once we’ve made further improvements with our DSG/DSP cross-linking procedure. As noted in the text, standard formaldehyde cross-linking didn’t give reproducible results for Elba1/Elba2, even though both should be directly contacting DNA. In order to get reproducible results for these two proteins, we had to use chemical cross-linking reagents like DSG/DSP, which have relatively long spacers between the two functional groups. Since Elba3 probably doesn’t contact the DNA at all, we expect that the ChIP experiments for this protein will be more difficult than for Elba1/Elba2.